# Inhibition of IRGM establishes a robust antiviral immune state to restrict pathogenic viruses

Parej Nath[1,2,†], Nishant Ranjan Chauhan[1,†] (ID), Kautilya Kumar Jena[1,†], Ankita Datey[3],
Nilima Dinesh Kumar[4] (ID), Subhash Mehto[1], Saikat De[3], Tapas Kumar Nayak[3], Swatismita Priyadarsini[1],
Kshitish Rout[1], Ramyasingh Bal[1], Krushna C Murmu[5] (ID), Manjula Kalia[6] (ID), Srinivas Patnaik[2],
Punit Prasad[5], Fulvio Reggiori[4] (ID), Soma Chattopadhyay[3,*] & Santosh Chauhan[1,**] (ID)

## Abstract

The type I interferon (IFN) response is the major host arsenal against invading viruses. IRGM is a negative regulator of IFN responses under basal conditions. However, the role of human IRGM during viral infection has remained unclear. In this study, we show that IRGM expression is increased upon viral infection. IFN responses induced by viral PAMPs are negatively regulated by IRGM. Conversely, IRGM depletion results in a robust induction of key viral restriction factors including IFITMs, APOBECs, SAMHD1, tetherin, viperin, and HERC5/6. Additionally, antiviral processes such as MHC-I antigen presentation and stress granule signaling are enhanced in IRGM-deficient cells, indicating a robust cell-intrinsic antiviral immune state. Consistently, IRGM-depleted cells are resistant to the infection with seven viruses from five different families, including *Togaviridae*, *Herpesviridae*, *Flaviviverdae*, *Rhabdoviridae*, and *Coronaviridae*. Moreover, we show that *Irgm1* knockout mice are highly resistant to chikungunya virus (CHIKV) infection. Altogether, our work highlights IRGM as a broad therapeutic target to promote defense against a large number of human viruses, including SARS-CoV-2, CHIKV, and Zika virus.

**Keywords** IRGM; IRGM1; CHIKV; ZIKV; SARS-CoV-2
**Subject Categories** Immunology; Microbiology, Virology & Host Pathogen Interaction; Signal Transduction

## Introduction

Deadly outbreaks of viruses are of significant human health concern. A large number of viral infections are treated using antiviral drugs, which typically target mechanisms of viral replication (Strasfeld & Chou, 2010). In most cases, due to selective pressure, the virus evolves faster to circumvent the targeted mechanism and become resistant to the drug (Strasfeld & Chou, 2010). In addition, vaccine efficacy is a significant problem in effectively restricting viral diseases because of the continuous evolution of viruses (Lipsitch, 2019). Due to these problems, the development of host-directed therapies for improving broad antiviral immunity could be a more effective approach, especially against emerging strains and viruses (Kaufmann *et al*, 2018). Understanding the mechanisms of antiviral host response is therefore crucial to identify new targets for host-based therapies.

The type I interferon (IFN) response is one of the robust antiviral components of the host's innate immune systems, which also operates cell autonomously (Teijaro, 2016). This is why a multitude of viruses have developed strategies to downregulate type I IFN response. The pattern recognition receptors (PRRs) such as cGAS, RIG-I, MDA5, TLR9, TLR7, and TLR3 sense nucleic acids pathogen-associated molecular patterns (PAMPs) of viral origin and induce downstream signaling events leading to the production of type I IFNs (McNab *et al*, 2015). The interaction of type I IFNs with cognate receptors activates JAK-STAT1/2 signaling leading to the transcriptional upregulation of hundreds of interferon-stimulated genes (ISG's) (Schoggins & Rice, 2011; McNab *et al*, 2015; Teijaro, 2016). The ISGs are the effector molecules of the IFN response and can autonomously inhibit every step of viral propagation, including virus cell entry, replication, transcription, translation, egression, and cell-to-cell transmission (Schoggins & Rice, 2011; Schoggins, 2019). In addition to the direct role of ISGs in viral inhibition, ISGs

1 Cell Biology and Infectious Diseases Unit, Department of Infectious Disease Biology, Institute of Life Sciences, Bhubaneswar, India
2 School of Biotechnology, KIIT University, Bhubaneswar, India
3 Molecular Virology Lab, Department of Infectious Disease Biology, Institute of Life Sciences, Bhubaneswar, India
4 Department of Biomedical Sciences of Cells and Systems, University of Groningen, University Medical Center Groningen, Groningen, The Netherlands
5 Epigenetic and Chromatin Biology Unit, Institute of Life Sciences, Bhubaneswar, India
6 Virology Lab, Regional Centre for Biotechnology, NCR Biotech Science Cluster, Faridabad, India
*Corresponding author. Tel: +91 0674 2304235; E-mail: soma@ils.res.in
**Corresponding author (lead contact). Tel: +91 0674 2304334; E-mail: schauhan@ils.res.in
†These authors contributed equally to this work

and IFNs have immunomodulatory functions (Schoggins & Rice, 2011; McNab *et al*, 2015; Schoggins, 2019). Several ISGs are potent chemokines and cytokines that trigger infiltration and activation of immune cells for clearance of infected cells (Schoggins & Rice, 2011; McNab *et al*, 2015; Schoggins, 2019). Thus, identification of gene targets that can induce host type I IFN response could be valuable for the development of prophylactic, host-based antiviral therapies.

Immunity-related GTPase M (IRGM) and its mouse orthologue Irgm1 are key negative regulators of inflammation (Bafica *et al*, 2007; Chauhan *et al*, 2015; Pei *et al*, 2017; Mehto *et al*, 2019a, 2019b). We and others have shown that IRGM negatively regulates the activation of NLRP3 inflammasomes to control colon inflammation in a colitis mouse model (Pei *et al*, 2017; Mehto *et al*, 2019a,b). Our recent work demonstrated that in basal conditions, IRGM is also a master negative regulator of type I IFN response by negatively regulating cGAS-STING, RIG-I-MAVS, and TLR3 signaling pathways (Jena *et al*, 2020). However, it is unclear how IRGM is regulated and how it modulates IFN response during viral infections. More importantly, it remains to be established whether inhibiting IRGM expression could induce broad antiviral immunity to new emerging viruses.

In this study, we show that a large number of well-established viral restriction factors and other antiviral mechanisms are induced upon IRGM depletion. Remarkably, inhibiting IRGM in human cells makes them resistant and/or resilient to the infection of human DNA and RNA viruses, including herpes simplex virus 1 (HSV-1), Zika virus (ZIKV), West Nile virus (WNV), CHIKV, vesicular stomatitis virus (VSV), Japanese encephalitis virus (JEV), and severe acute respiratory syndrome coronavirus 2 (SARS-CoV-2 virus). In line with these findings, IRGM knockout mice were highly resistant to CHIKV infection. Taken together, this work establishes that blocking IRGM expression could be an effective approach to induce broad antiviral immunity and protect individuals from infections caused by life-threatening viruses.

# Results and discussion

### Viruses and viral PAMPs induce IRGM expression

First, we tested whether IRGM expression is modulated by a viral infection and synthetic analogs of viral PAMPs. We found that infection of three different viruses including CHIKV, JEV, and HSV-I increased the IRGM protein expression (Fig 1A–C). The synthetic analogs of viral PAMPs such as polyinosinic:polycytidylic acid (poly (I:C), a synthetic analog of dsRNA), triple phosphate RNA (5'pppRNA, a synthetic ligand of RIG-I), and poly (deoxyadenylic-deoxythymidylic) acid (poly (dA-dT), a synthetic analog of B-DNA) also significantly enhanced the expression of IRGM at mRNA and protein levels (Figs 1D–F, and EV1A and B). During DNA virus infection, the cytosolic dsDNA is detected by cGAS leading to the synthesis of a second messenger cyclic GMP-AMP (cGAMP). cGAMP induces the STING pathway, which in turn increases type I IFN response and an antiviral state (Ni *et al*, 2018). We found that both dsDNA and cGAMP robustly induced the expression of IRGM (Figs 1G and H, and EV1C). Altogether, these data show that IRGM expression is increased by viral infections and PAMP treatment.

### Viral PAMP-induced type I IFN response is negatively regulated by IRGM

Viral PAMPs such as poly (I:C) and poly (dA-dT) are robust inducers of type I IFN response. To test how IRGM modulates viral PAMP-induced IFN response, we first employed a THP-1 luciferase IFN reporter cell line (InvivoGen). The poly (I:C)- and poly (dA-dT)-induced IFN response was substantially increased upon *IRGM* knockdown (Fig 1I and J). A similar effect was observed upon exposure of cGAMP- and interferon-β (IFN-β) (Fig 1K and L). The *IRGM* knockdown efficiency was ~75–80% in THP-1 cells (Fig EV1D). To further substantiate the finding, we performed qRT–PCR with the *IFN-β* gene and the sentinel IFN responsive genes. The poly (I:C)- and poly (dA-dT)-induced expression of *IFN-β*, *MX2*, *ISG15*, and *OAS1* was strongly increased upon IRGM knockdown (Fig 1M–T). Taken together, the data suggest that the viral PAMPs-induced IFN response is negatively regulated by IRGM.

### Several classical cell-autonomous viral restriction factors are upregulated in IRGM-depleted cells

Viral restriction factors are host cellular proteins that establish the first line of defense and are capable of blocking almost all stages of the viral life cycle, including viral entry, replication, assembly, and egress/release (Goff, 2004; Colomer-Lluch *et al*, 2018; Urbano *et al*, 2018; Chemudupati *et al*, 2019; Boso & Kozak, 2020) (Fig 2A). Each of these factors is self-sufficient in conferring an effective and early restriction of viruses. The classical host antiviral restriction factors that are known for their activity against a large number of RNA and

---

**Figure 1. Viruses and viral PAMP-induced IRGM expression suppress IFN response.**

A–C  Western blot analysis of THP-1 cells infected with (A) HSV-1 at MOI 5 for 3 h, 6 h, 12 h, 18 h, and 24 h (B) CHIKV at MOI 5 for 3 h, 6 h, 12 h, 18 h, and 24 h (C) JEV at MOI 5 for 3 h, 6 h, 12 h, and 24 h.

D–H  Western blot analysis of THP-1 cells untreated or treated with (D) 5'pppRNA (1 μg/ml) for 3 h, 6 h, 12 h, 18 h and 24 h (E) Poly I:C (1 μg/ml) for 3 h, 6 h, 9 h, 12 h and 24 h (F) Poly dA:dT (1 μg/ml) for 3 h, 6 h, 9 h, 12 h and 24 h (G) cGAMP (1 μg/ml) for 6 h, 12 h, 18 h, and 24 h (H) dsDNA (1 μg/ml) for 3 h, 6 h, and 12 h and probed with antibodies as indicated.

I–L  Control and IRGM knockdown THP-1 IFN reporter cells were treated with (I) Poly I:C (1 μg/ml) for 8 h, 12 h and 24 h or (J) Poly dA:dT (1 μg/ml) for 12 h and 24 h or (K) IFN-β (500 ng/ml) for 8 h and 12 h or (L) cGAMP (1 μg/ml) for 12 h and 24 h and the supernatant was subjected to luciferase reporter assay using QUANTI-Luc reagent (*n* = 3, mean ± SD, **P* < 0.05, ***P* < 0.005, ****P* < 0.0005, Student's unpaired *t*-test).

M–P  Control and IRGM knockdown THP-1 cells were untreated and treated with Poly I:C (1 μg/ml) for 8 h and the total RNA was subjected to qRT–PCR with primers of (M) MX2 (N) ISG15 (O) OAS1 and (P) IFN-β. (*n* = 3, mean ± SD, **P* ≤ 0.05, ***P* ≤ 0.005, ****P* < 0.0005, Student's unpaired *t*-test).

Q–T  Control and IRGM knockdown THP-1 cells were untreated and treated with Poly dA:dT (1 μg/ml) for 8 h and the total RNA was subjected to qRT–PCR with primers of (Q) MX2 (R) ISG15 (S) OAS1 and (T) IFN-β. (*n* = 3, mean ± SD, ***P* ≤ 0.005, ****P* < 0.0005, Student's unpaired *t*-test).

Source data are available online for this figure.

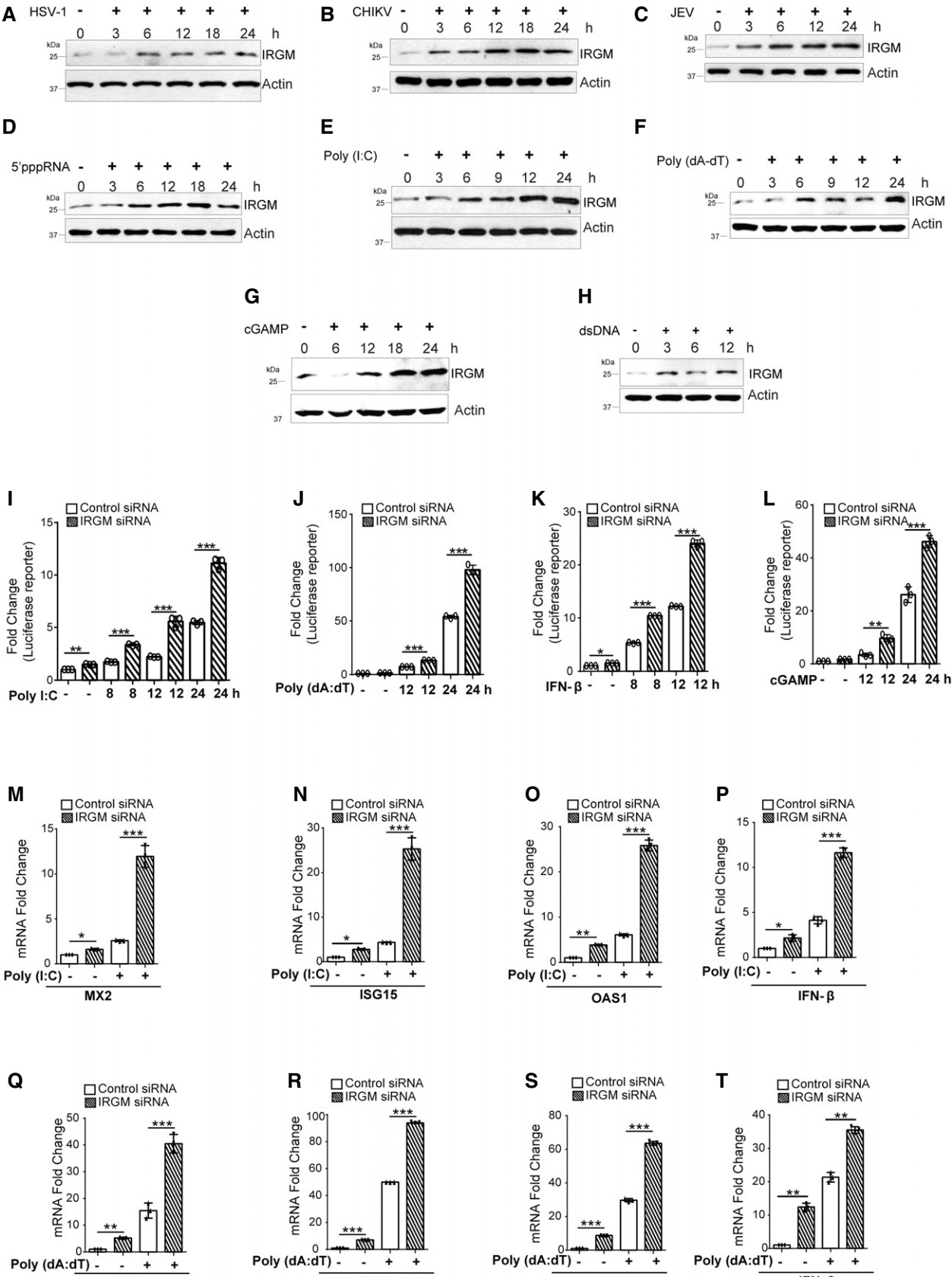

**Figure 1.**

DNA viruses are IFITMs, APOBECs, SAMHD1, SAMD9L, tetherin/ BST2, RSAD2/ viperin, HERC5/6, OAS's, MX1/2, ISG15, PKR, and TRIM5α (Goff, 2004; Colomer-Lluch *et al*, 2018; Urbano *et al*, 2018; Chemudupati *et al*, 2019; Boso & Kozak, 2020) (Fig 2A). Interferon-induced transmembrane proteins (IFITMs) localize in the cell membranes and restrict virus infection at cell entry by interrupting the membrane fusion between the viral envelope and cellular membranes (Weidner *et al*, 2010; Li *et al*, 2013; Perreira *et al*, 2013; Compton *et al*, 2014). SAM domain and HD domain-containing protein 1 (SAMHD1), apolipoprotein B mRNA editing enzyme3 (APOBEC3), and viperin (RSAD2) are nucleotide manipulating enzymes and restrict a variety of viruses by manipulating host NTP's/dNTP's or viral nucleic acid (Sze *et al*, 2013; Gizzi *et al*, 2018; Coggins *et al*, 2020). IFIT's can inhibit mRNA translation initiation by binding to the multisubunit eukaryotic translation initiation factor 3 (eIF3) and interfering with the assembly of the preinitiation complex and can inhibit specifically translation of viral mRNA (Fensterl & Sen, 2015). Human HERC5 and HerC6 inhibit virus particle assembly by conjugating ISG15 (ISGylation) to various viral proteins (Durfee *et al*, 2010; Oudshoorn *et al*, 2012). Tetherin broadly restricts enveloped virus release by tethering budded viral particles to the plasma membrane (Neil, 2013). We compared the transcriptome (Jena *et al*, 2020) of control and IRGM-deficient cells to determine whether the expression of these factors is changed in the absence of IRGM. To our surprise, almost all of the established classical host restriction factors were significantly ($P$-value < 0.05, > 1.5 folds) upregulated in RNA sequencing analysis of stable IRGM knockdown HT-29 epithelial cells (Fig 2B) and bone marrow-derived macrophages (BMDMs) from *Irgm1*$^{-/-}$ knockout mice (Fig 2C).

To confirm the results of RNA sequencing, we performed qRT–PCR for key antiviral restriction factors using RNA from mouse BMDMs and human cell lines. All of the tested antiviral restriction factors were induced in IRGM knockdown HT-29 cells and *Irgm1*$^{-/-}$ BMDMs (Figs 2D, and EV2A and B). The levels of IFNβ in serum of *Irgm1*$^{-/-}$ knockout mice were found to be significantly higher than wild-type mice (Fig EV2C). Next, we analyzed the protein levels of several viral restriction factors, including IFITM3, SAMHD1, APOBEC3G, viperin, ISG15, and tetherin. All of these viral restriction factors were dramatically induced in IRGM knockdown HT-29 cells compared to control cells (Fig 2E). Also, enhanced ISGylation of proteins, a hallmark of viral infection, was observed in IRGM-depleted cells (Fig 2E). On contrary, the overexpression of IRGM suppressed the levels of viral restriction factors (Fig EV2D).

To ensure the specificity of the IRGM-mediated regulation, we performed a rescue experiment by complementing IRGM in IRGM knockdown cells. The increased levels of viral restriction factors in IRGM-depleted cells were restored to basal levels in the complemented cells (Figs 2F and EV2E). The GTPase activity of IRGM is essential for its autophagy and anti-inflammatory functions (Singh *et al*, 2006; Kumar *et al*, 2018; Mehto *et al*, 2019b). The conversion of amino acid residue serine to asparagine at 47$^{th}$ position (S47N mutation) renders IRGM inactive (Singh *et al*, 2006; Kumar *et al*, 2018; Mehto *et al*, 2019b). We compared wild-type IRGM and S47N mutant for their ability to regulate the expression levels of viral restriction factors. As compared to wild type, the catalytic mutant of IRGM was not able to efficiently suppress the expression of viral restriction factors (Fig 2G) suggesting that GTPase activity of IRGM is required for suppression of IFN response.

Taken together, the data show that a large number of the classical host viral restriction factors are robustly induced upon depletion of IRGM in human and mouse cells.

---

**Figure 2.  Depletion of IRGM induces key viral restriction factors and antiviral mechanisms.**

A  Pictorial representation of stages (black font) of a typical life cycle of RNA viruses with host viral restriction factors (red font) induced in IRGM/Irgm1 depleted cells. Created using Biorender.com.

B, C  Heatmaps representing the expression pattern of viral restriction factors from RNA sequencing data in (B) control and IRGM knockdown shRNA stable HT-29 cells (3 biological replicates) and (C) *irgm1*$^{+/+}$ and *irgm1*$^{-/-}$ mice BMDMs (2 biological replicates).

D  RNA was isolated from *irgm1*$^{+/+}$ and *irgm1*$^{-/-}$ BMDMs and subjected to qRT–PCR with indicated viral restriction factor genes ($n = 3$, mean ± SD, *$P < 0.05$, **$P \leq 0.005$, ***$P < 0.0005$, Student's unpaired $t$-test).

E  Western blot analysis with lysates of control and *IRGM*$^{+/-}$ HT-29 cells with indicated antibodies of viral restriction factors.

F  Western blot analysis with lysates of control or *IRGM*$^{+/-}$ HT-29 cells or Flag IRGM complemented *IRGM*$^{+/-}$ HT-29 cells with indicated antibodies of viral restriction factors.

G  RNA isolated from control or Flag IRGM transfected or Flag IRGM S47N transfected THP-1 cells subjected to qRT–PCR with indicated genes. ($n = 3$, mean ± SD, **$P < 0.005$, ***$P < 0.0005$, Student's unpaired $t$-test).

H–J  RNA isolated from control and IRGM knockdown HT-29 cells and subjected to qRT–PCR with indicated genes of (H) Immunoproteasome complex (I) TAP complex (J) human leukocyte antigen (HLA) system. ($n = 3$, mean ± SD, *$P < 0.05$, **$P < 0.005$, ***$P < 0.0005$, Student's unpaired $t$-test).

K  Antigen uptake assay shown by representative flow cytometry analysis of control and IRGM siRNA transfected THP-1 cells treated with OVA conjugate AF488 (5 µg/ml, 30 min). The graph depicts the mean fluorescence intensity of control and IRGM knockdown THP-1 cells treated with OVA conjugate AF488 ($n = 3$, mean ± SD, **$P < 0.005$, Student's unpaired $t$-test).

L, M  Antigen processing assay shown by representative confocal images of control and IRGM siRNA transfected THP-1 cells treated with DQ-OVA (green) (10 µg/ml, 30 min). Scale Bar, 10 µm. (M) Graph depicts percentage of control and IRGM knockdown THP-1 cells with DQ-OVA puncta's ($n = 3$, mean ± SD, ***$P < 0.0005$, Student's unpaired $t$-test).

N, O  SIINFEKL based Antigen presentation assay shown by representative flow cytometry analysis of H-2K$^b$-SIINFEKL on the surface of *Irgm1*$^{+/+}$ and *irgm1*$^{-/-}$ mouse BMDMs treated with OVA (2 mg/ml, 3 h). (O) The graph depicts the mean fluorescence intensity of H-2K$^b$- SIINFEKL on the surface of *Irgm1*$^{+/+}$ and *irgm1*$^{-/-}$ mouse BMDMs treated with OVA (2 mg/ml, 3 h) ($n = 3$, mean ± SE, **$P < 0.005$, Student's unpaired $t$-test).

P  Representative confocal images of H-2K$^b$-SIINFEKL (red) on the surface of *Irgm1*$^{+/+}$ and *irgm1*$^{-/-}$ mouse BMDMs treated with OVA (2 mg/ml, 3 h), Scale bar 5 µm or 8 µm (as indicated).

Q  Western blot analysis with lysates of control and IRGM siRNA knockdown HT-29 cells with indicated antibodies of stress granules signaling pathway.

R  Representative Immunofluorescence confocal images of *irgm1*$^{+/+}$ and *irgm1*$^{-/-}$ mouse BMDMs immunostained with dsRNA (green) and TIA-1 (red) antibody. Scale bar 3 µm.

Source data are available online for this figure.

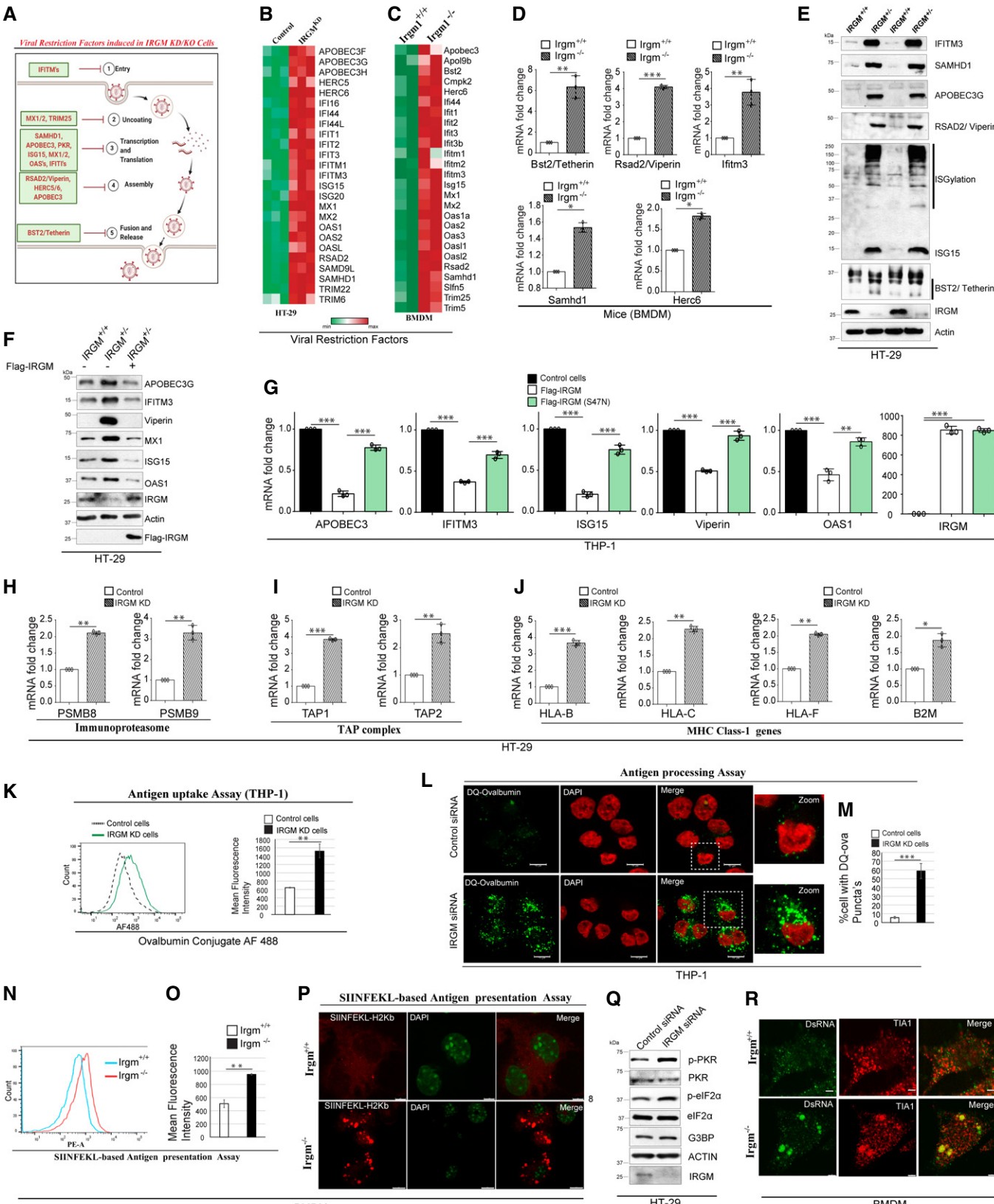

**Figure 2.**

## Antigen presentation pathways and stress granule PKR signaling are activated in IRGM-depleted cells

We have previously shown that IRGM depletion induces the expression of genes involved in the MHC-mediated antigen presentation pathways (Jena *et al*, 2020). However, we did not test whether this induction leads to an effective upregulation of antigen uptake, processing, and presentation. While the MHC class II pathway presents exogenous antigens via the endolysosomal pathway, the MHC class I pathway presents endogenous peptides derived from viral antigens, via the immunoproteasome (Fig EV2F) (Hewitt, 2003). MHC class I molecules, in complex with β2-Microglobulin

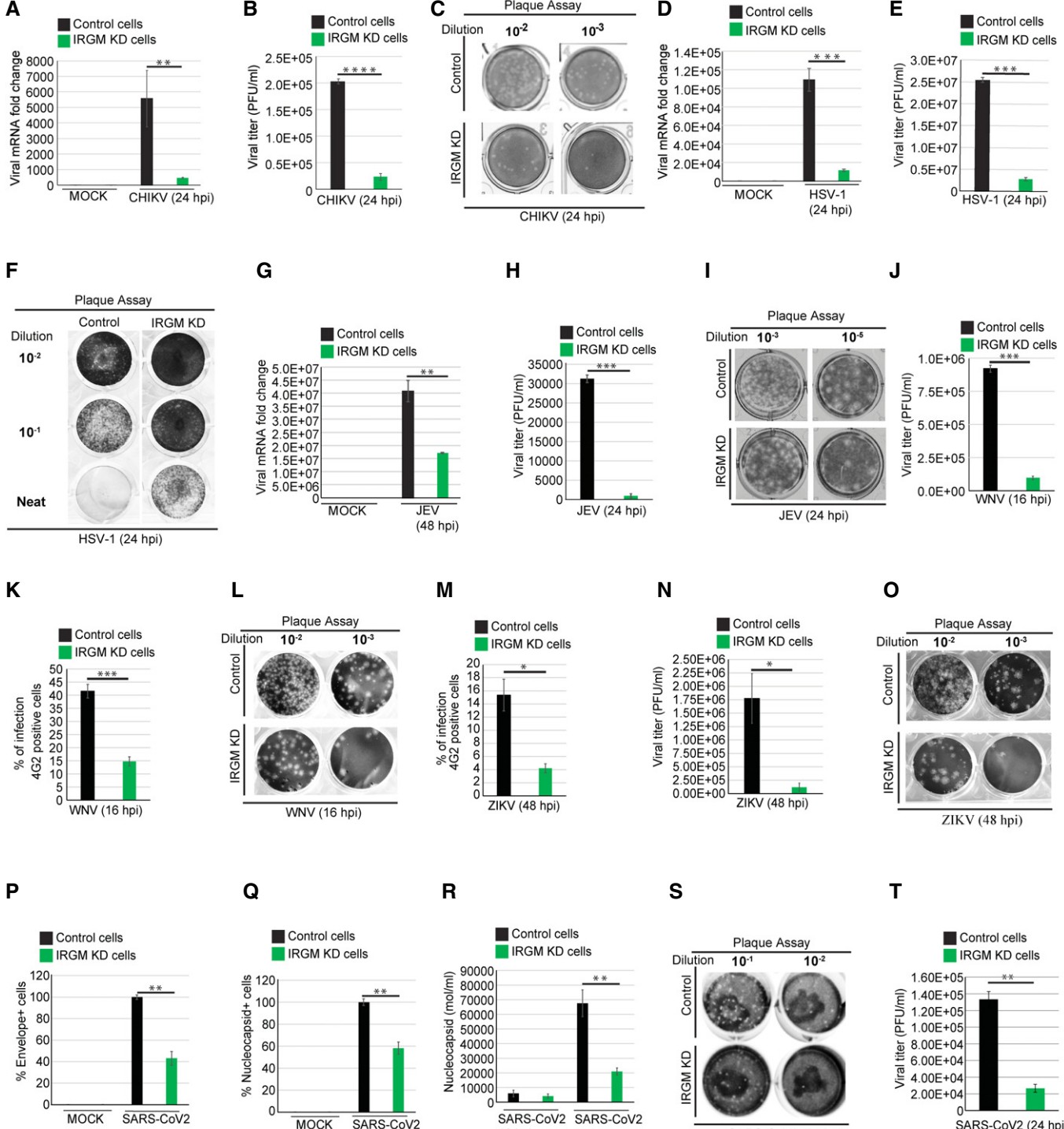

**Figure 3.**

◀ **Figure 3. IRGM-depleted cells can restrict infection with CHIKV, HSV-1, JEV, VSV, ZIKV, WNV, and SARS-CoV-2.**

A   Total RNA was isolated from mock and CHIKV (MOI 1, 24 h) infected control and IRGM knockdown HT-29 cells and subjected to qRT–PCR with CHIKV specific primers to quantitate total viral load ($n = 3$, mean ± SE, **$P < 0.005$, Student's unpaired $t$-test).

B   The graph depicts quantification of CHIKV plaque assays (plaque-forming units/ml) in Vero cells performed from the culture supernatant of CHIKV (MOI 1, 24 h) infected HT-29 control and IRGM knockdown cell ($n = 3$, mean ± SE, ****$P < 0.00005$, Student's unpaired $t$-test).

C   Representative images of the plaque assay in Vero cells performed from the culture supernatant of CHIKV (MOI 1, 24 h) infected HT-29 control and IRGM knockdown cells.

D   Total RNA was isolated from mock and HSV-1 (MOI 1, 24 h) infected control and IRGM knockdown HT-29 cells and subjected to qRT–PCR with HSV-1 specific primers to quantitate total viral load ($n = 3$, mean ± SE, ***$P < 0.0005$, Student's unpaired $t$-test).

E   The graph depicts quantification of HSV-1 plaque assays (plaque-forming unit/ml) in Vero cells performed from culture supernatant of HSV-1 (MOI 1, 24 h) infected HT-29 control and IRGM knockdown cell ($n = 3$, mean ± SE, ***$P < 0.0005$, Student's unpaired $t$-test).

F   Representative images of plaque assay in Vero cells performed from the culture supernatant of HSV-1 (MOI 1, 24 h) infected HT-29 control and IRGM knockdown cells.

G   Total RNA was isolated from mock and JEV (MOI 1, 48 h) infected control and IRGM knockdown HT-29 cells subjected to qRT–PCR with JEV specific primers to quantitate total viral load ($n = 3$, mean ± SE, **$P < 0.005$, Student's unpaired $t$-test).

H   The graph depicts quantification of JEV plaque assays (plaque-forming unit/ml) in Vero cells performed from the culture supernatant of JEV (MOI 1, 24 h) infected HT-29 control and IRGM knockdown cell ($n = 3$, mean ± SE, ***$P < 0.0005$, Student's unpaired $t$-test).

I   Representative images of plaque assay in Vero cells performed from the culture supernatant of JEV (MOI 1, 24 h) infected HT-29 control and IRGM knockdown cells.

J   The graph depicts the percentage of WNV (MOI 0.3, 16 h) infected control and IRGM knockdown Huh7 cells stained with 4G2 antibody analyzed by flow cytometry ($n = 3$, mean ± SE, ***$P < 0.0005$, Student's unpaired $t$-test).

K   The graph depicts quantification of WNV plaque assays (plaque-forming units/ml) in Vero cells performed from culture supernatant of WNV (MOI 0.3, 16 h) infected Huh7 control and IRGM knockdown cell ($n = 3$, mean ± SE, ***$P < 0.0005$, Student's unpaired $t$-test).

L   Representative images of plaque assay in Vero cells were performed from the culture supernatant of WNV (MOI 0.3, 16 h) infected HuH7 control and IRGM knockdown cells.

M   The graph depicts the percentage of ZIKV (MOI 5, 48 h) infected control and IRGM knockdown HuH7 cells stained with 4G2 antibody analyzed by flow cytometry ($n = 3$, mean ± SD, *$P < 0.05$, Student's unpaired $t$-test).

N   The graph depicts quantification of ZIKV plaque assays (plaque-forming units/ml) in Vero cells performed from the culture supernatant of ZIKV (MOI 5, 48 h) infected Huh7 control and IRGM knockdown cells ($n = 3$, mean ± SD, *$P < 0.05$, Student's unpaired $t$-test).

O   Representative images of plaque assay in Vero cells were performed from the culture supernatant of ZIKV (MOI 5, 48 h) infected Huh7 control and IRGM knockdown cells.

P   Total RNA was isolated from mock and SARS-CoV-2 (MOI 1, 24 h) infected control and IRGM knockdown THP-1 cells and subjected to qRT–PCR with envelope specific primers of SARS-CoV-2 to quantitate total viral load ($n = 3$, mean ± SE, **$P < 0.005$, Student's unpaired $t$-test).

Q   Total RNA was isolated from mock and SARS-CoV-2 (MOI 1, 24 h) infected control and IRGM knockdown THP-1 cells and subjected to qRT–PCR with nucleocapsid specific primers of SARS-CoV-2 to quantitate total viral load ($n = 3$, mean ± SE, **$P < 0.005$, Student's unpaired $t$-test).

R   Total RNA was isolated from the supernatant of mock and SARS-CoV-2 (MOI 1, 10 h and 20 h) infected control and IRGM knockdown THP-1 cells and subjected to qRT–PCR with nucleocapsid specific primers of SARS-CoV-2 to quantitate total viral load ($n = 3$, mean ± SE, **$P < 0.005$, Student's unpaired $t$-test).

S   Representative images of plaque assay in Vero E6 cells were performed from the culture supernatant of SARS-CoV-2 (MOI 1, 24 h) infected THP-1 control and IRGM knockdown cells.

T   The graph depicts quantification of SARS-CoV-2 plaque assays (plaque-forming units/ml) in Vero E6 cells performed from the culture supernatant of SARS-CoV-2 (MOI 1, 24 h) infected THP-1 control and IRGM knockdown cells ($n = 3$, mean ± SE, **$P < 0.005$, Student's unpaired $t$-test).

(B2M), are loaded with endogenous peptides that are generated by the immunoproteasome (PSMB8 and PSMB9) and subsequently imported into the ER by the heterodimeric TAP1/TAP2 transporter. The depletion of IRGM resulted in the induction of classical (HLA-B and HLA-C) and non-classical (HLA-F) MHC class I molecules, B2M, PSMB8/9, and TAP1/2 genes (Fig 2H–J). Several of these MHC-1 molecules are shown to be regulated by NLRC5 (Meissner *et al*, 2010; Kobayashi & van den Elsen, 2012; Neerincx *et al*, 2012), which itself is an ISG and is regulated by IRGM (Jena *et al*, 2020).

We also performed functional assays to understand whether IRGM-depleted cells have increased capacity to uptake and process antigens. Alexa Fluor 488-conjugated ovalbumin (OVA) is commonly used to study the uptake of antigens (Burgdorf *et al*, 2007). Indeed, IRGM knockdown THP-1 cells showed a significantly higher ability to uptake Alexa Fluor 488-conjugated OVA from the medium (Figs 2K and EV2G). Next, we used DQ-OVA to test the antigen uptake and processing capacity of the same cells. DQ-OVA is a self-quenched conjugate of OVA and only exhibits green fluorescence upon proteolytic degradation. Thus, it is extensively used to study antigen processing (Wahid *et al*, 2005; von Garnier *et al*, 2007). We detected a significantly large number of cells displaying fluorescent puncta in IRGM knockdown THP-1 macrophages in comparison to the control (Fig 2L and M) suggesting an increased ability of antigen uptake and processing in these cells. To test whether this phenomenon is specific for OVA or general endocytosis is induced in IRGM-depleted cells, we monitored the uptake of transferrin, a protein whose internalization in cells is dependent on receptor-mediated endocytosis. We found increased internalization of transferrin in IRGM knockdown cells (Fig EV2H) suggesting that IRGM depletion enhances endocytosis *per se*.

Next, we utilized an extensively used OVA-derived SIINFEKL-peptide-based system to measure MHC Class I antigen presentation efficiency (Dersh *et al*, 2019). The 25-D1.16 monoclonal antibody reacts with the SIINFEKL peptide bound to H-2Kb of MHC class I, but not with unbound H-2Kb, or H-2Kb loaded with an irrelevant peptide. BMDMs from *Irgm1*$^{+/+}$ and *irgm1*$^{-/-}$ mice were pulsed with OVA before staining them with phycoerythrin (PE)-conjugated 25-D1.16 monoclonal antibody for FACS and microscopic analysis. Both assays showed a considerably higher ability of *irgm1*$^{-/-}$ BMDMs to process OVA and present the SIINFEKL peptide with MHC class I molecule (Figs 2N–P and EV2I). Taken together, these results show that IRGM-depleted cells have an augmented capacity in antigen uptake, processing, and presentation by the MHC class I system.

Viral-sensing activates IFN-inducible RNA-dependent protein kinase (PKR/EIF2AK2 gene), and the downstream phosphorylation of eukaryotic translation initiation factor 2 alpha (eIF2α) leads to both the inhibition of host protein translation and the formation of cytoplasmic stress granules (McCormick & Khaperskyy, 2017). Stress granules strongly inhibit viral replication (McCormick & Khaperskyy, 2017), and therefore, several viruses have evolved mechanisms to suppress stress granule formation. The phosphorylation of eIF2α by PKR is sufficient to induce stress granule formation even in uninfected cells. We found increased phosphorylation of PKR and eIF2α in IRGM knockdown cells (Fig 2Q), whereas no significant change in G3BP protein levels was observed. Importantly, very large stress granules-like structures, identified using the stress granule marker TIA-1 and dsRNA antibody, were observed in $irgm1^{-/-}$ BMDMs but not in the control cells (Fig 2R). The induction of stress granule signaling in IRGM-deficient cells is probably due to the enhanced IFN response, which can lead to the activation of PKR and its downstream signaling. Altogether, the data suggest that multiple host antiviral mechanisms are induced in IRGM-depleted cells.

## IRGM-depleted cells are resistant to numerous viral infections including ZIKV and SARS-CoV-2

The above data suggest that blocking IRGM expression may provide an inimical environment for the virus propagation in host cells. We tested this hypothesis by infecting cell lines of epithelial (Huh7 and HT-29) and monocytic (THP-1) origin with viruses from different families. First, we assessed the replication of CHIKV in control and IRGM knockdown HT-29 cells.

CHIKV is a mosquito-transmitted positive-sense ssRNA virus belonging to the *Togaviridae* family. We found that CHIKV replication was significantly reduced (80–90%, 24 hpi) in IRGM-deficient cells (Fig 3A). Consistently, there was a strong reduction (10 fold) in total infectious virus particles produced by IRGM knockdown cells compared to the control in plaque assays (Fig 3B and C). Further, in agreement, replication of CHIKV was found to be significantly higher in IRGM overexpressing cells (18 hpi) (Fig EV3A).

HSV-1 is a sexually transmitted double-stranded DNA virus from the *Herpesviridae* family. We observed a significant inhibition (~ 90% at 24 hpi) of HSV-1 replication in IRGM-depleted HT-29 cells compared to the control cells (Fig 3D). Measurement of produced infectious HSV-1 particles using the plaque assay also showed a dramatic reduction in IRGM-deficient cells in comparison to the controls (Fig 3E and F).

We next tested the replication of three positive-sense ssRNA viruses of the *Flaviviridae* family, JEV, ZIKV, and WNV. As compared to control cells, IRGM knockdown cells showed inhibition of JEV replication and infectious particle egression of > 50% and > 90%, respectively (Fig 3G–I). The replication of the ZIKV and WNV was compared in control and IRGM-depleted Huh7 cells. The percentage of infected cells was determined by flow cytometry after staining cells with primary antibody against protein E, a flavivirus group antigen. This analysis revealed that IRGM-depleted cells could robustly restrict the growth of the ZIKV and WNV (Fig 3J and M). In plaque assays, more than 90% inhibition of WNV (Fig 3K and L) and ZIKV virus was observed in IRGM knockdown cells (Fig 3N and O).

We also tested replication of VSV, a negative sense ssRNA virus from the *Rhabdoviridae* family that mainly infects animals and is the most common virus used in the laboratory to study immune responses. Strong inhibition of VSV replication was observed in IRGM knockdown HeLa cells compared to the control cells (Fig EV3B). We exposed the control and IRGM knockdown HeLa and THP-1 cells to a VSV strain expressing GFP for 4 h. The GFP fluorescence was first observed in control cells as compared to IRGM knockdown cells and also a lower fluorescence was observed in IRGM knockdown HeLa and THP-1 cells (Fig EV3C) indicating a reduced propagation of the virus. Further, we tested whether the complementation of IRGM in IRGM knockdown cells can rescue the viral replication defect. Indeed, the viral replication in Flag IRGM complemented IRGM knockdown cells was similar to wild-type cells (Fig EV3D).

The ongoing SARS-CoV-2 outbreak has posed an enormous threat to global public health. SARS-CoV-2 is a positive-sense ssRNA virus belonging to the *Coronaviridae* family. A robust induction of type I IFN response upon SARS-CoV-2 infection is reported (Winkler *et al*, 2020; Zhou *et al*, 2020). Moreover, like other viruses, SARS-CoV-2 infection can be inhibited by the IFN response (Lokugamage *et al*, 2020; Mantlo *et al*, 2020). Also, inborn errors of type I IFN immunity or the presence of autoantibodies against type I IFNs in patients can lead to life-threatening COVID-19 (Bastard *et al*, 2020; Meffre & Iwasaki, 2020; Zhang *et al*, 2020b). Further, it is demonstrated that administration of IFN-I pre- or post-virus challenge can suppress SARS-CoV-2 infection (Hoagland *et al*, 2021). Taken together, these studies suggest that upregulating IFN response could be an important prophylactic measure to restrict SARS-CoV-2 infection. Since IRGM depletion results in constitutive upregulation of IFN response, this model permits to directly test the effect of an enhanced type I IFN response on SARS-CoV-2 infection.

First, we compared the transcriptome of IRGM knockout cells (Jena *et al*, 2020) to the transcriptomes (or proteomes) modulated by SARS-CoV-2 infection in seven different conditions from four different studies (Blanco-Melo *et al*, 2020; Wilk *et al*, 2020; Zhang *et al*, 2020a; Wyler *et al*, 2021) using metascape (https://metascape.org/COVID) (Zhou *et al*, 2019). There was a significant overlap in transcriptome induced in IRGM-deficient cells with the transcriptomes induced upon SARS-CoV-2 infection but not with the transcriptomes that were downregulated upon infection (Fig EV3E). A gene overlap analysis using circos plot showed a high degree of overlap between genes upregulated in IRGM-deficient cells and genes induced upon SARS-CoV-2 infection in different cell types and different studies (Fig EV3F). The Gene Ontology (GO) enrichment analysis comparing a few SARS-CoV-2 induced transcriptomes with IRGM$^{-/-}$ transcriptome includes GO terms like "positive regulation of cytokine", "Interferon response," and "Defense response to virus" that are related to the activated immune system (Fig EV3G). This analysis suggests that the host immune defense that is raised against infection of SARS-CoV-2 is already possessed by IRGM-deficient cells.

Previous studies suggest that THP-1 is permissive to SARS-CoV and other coronavirus infections (Ng *et al*, 2004; Yen *et al*, 2006; Desforges *et al*, 2007). We infected THP-1 cells with SARS-CoV-2 for 1.5 h, washed, and then further incubated for either 10 or 20 h. We examined the presence of progeny virions in the supernatant by qRT–PCR. We observed > 8 folds induction in nucleocapsid RNA in

the supernatant of 20 hpi samples as compared to the 10 hpi (Fig EV3H), indicating that THP-1 is permissive for infection of SARS-CoV-2 as also suggested by a recent report (Boumaza *et al*, 2021). The plaque assays were also performed from the supernatant to confirm the production of infectious viral particles. The plaque formed was small and less in number (Fig EV3I) indicating that THP-1 cells are permissive for SARS-CoV-2 infection but less permissive for production of the virion.

Next, control and IRGM knockdown THP-1 cells were infected with SARS-CoV-2 for 24 h, and subsequently, the qRT–PCR analysis was performed from RNA isolated from cells and supernatant. The expression of the envelope and nucleocapsid genes was reduced by approximately 40–60% in IRGM knockdown cells compared to the control (Fig 3P and Q). Similar results were obtained with the culture supernatants (Fig 3R). Measurement of infectious SARS-CoV-2 particles produced using the plaque assay showed that IRGM knockdown cells were significantly resistant to the replication of SARS-CoV-2 in comparison to the controls (Fig 3S and T). The levels of ACE2 receptor in control and IRGM knockdown cells were found to be similar suggesting that differential expression of ACE2 does not contribute to lower viral load in IRGM-depleted cells (Fig EV3J). Altogether, the results show that IRGM-depleted cells efficiently suppress SARS-CoV-2 infection.

The data presented here strongly suggest that suppressing IRGM levels in cells can robustly restrict the infection of viruses belonging to different virus families.

### Irgm1$^{-/-}$ mice are resistant to CHIKV infection

Next, we employed a CHIKV infection neonatal mouse model (Couderc *et al*, 2008) to determine the role of IRGM *in vivo*. In this model, the young age and the inefficient type I IFN signaling have been found to be the reasons for severe disease outcomes (Couderc *et al*, 2008). Intradermal injection of $10^7$ plaque-forming units (PFU) of CHIKV to the 6 days old wild-type C57BL/6 neonates can induce chikungunya disease symptoms. Those are characterized by mild paralysis in one leg at 3 days post-infection (dpi), severe leg paralysis at 6–7 dpi, and death of the animals at 12–15 dpi (Fig 4A and Movie EV1) (Couderc *et al*, 2008). Accordingly, CHIKV-infected WT neonate mice displayed all of the progressive symptoms before they died (Fig 4A and Movie EV1). Even before the paralysis, the WT neonate mice appear to have problems including difficulty in maintaining body stability, seizures-like conditions, difficulty in walking, etc (Movie EV2).

CHIKV-infected *Irgm1$^{+/+}$* neonate mice displayed no increase in body weight until 6 dpi, which was then followed by a reduction in body weight until they died (Fig 4B). In contrast, a constant increase in body weight was observed in *irgm1$^{-/-}$* neonates until the termination of the experiments (Fig 4B). All of the twelve CHIKV-infected *Irgm1$^{+/+}$* neonate mice developed typical symptoms of paralysis by 5–6$^{th}$ dpi (Fig 4C). In striking contrast, there were no visible symptoms of paralysis in all the twelve *Irgm1$^{-/-}$* neonate mice subjected to the same infection by 5–6 dpi (Fig 4C and Movie EV3).

Out of twelve, ten of the wild-type mice died 15 dpi (point of termination of the experiment). In the case of *irgm1$^{-/-}$* neonates, two mice developed mild paralysis on day 10, and one died at 12 dpi (Fig 4D). None of the other ten *irgm1$^{-/-}$* neonate mice showed

any disease symptoms until the termination of the experiments (Fig 4D). This result reveals that a very strong intrinsic antiviral state exists in neonates of *irgm1$^{-/-}$* mice, which can prevent lethal CHIKV virus infection. We also measured the viral load in the muscles, liver, and brains of *Irgm1$^{+/+}$* and *irgm1$^{-/-}$* mice by estimating CHIKV infection by qRT–PCR (Fig 4E–G). We observed substantial viral load in both the muscles and the brain of the *Irgm1$^{+/+}$* mice, while CHIKV was practically undetectable in the same tissues from the *irgm1$^{-/-}$* animals. To understand the status of the IFN response, we also determined the expression of *MX2*. As expected, there was robust induction of *MX2* expression in the infected wild-type mice in comparison to the mock-treated animals (Fig 4H and I). The expression of *MX2* was also significantly increased in infected *irgm1$^{-/-}$* mice, but consistently with the symptoms, the induction was significantly less than infected *Irgm1$^{+/+}$* mice. Altogether, these results indicate that *irgm1$^{-/-}$* mice have the augmented capacity to restrict CHIKV infection.

Taken together, the *in vitro* and the *in vivo* results demonstrate that IRGM inhibition could be a potent therapeutic target for inducing a robust, broad-spectrum antiviral host immunity.

### Type I IFN response is blunted in virus-infected IRGM knockdown cells

The *irgm1$^{-/-}$* mice evoked a milder type I IFN response than the wild-type animals when exposed to CHIKV (Fig 4G–I). This result was somehow opposite to the one obtained with the synthetic viral PAMPs, where we found that the type I IFN response was further increased upon IRGM knockdown (Fig 1J–U). We hypothesized that this could be due to the capability of IRGM-deficient cells to efficiently block viral multiplication resulting in lower levels of PAMPs in the milieu.

To test this notion, we examined the status of the type I IFN response in cells in the presence or the absence of IRGM upon infection. In luciferase-based IFN reporter assays, CHIKV, JEV, HSV-1, and VSV induced IFN response was significantly less in IRGM-depleted cells (Fig 5A–D). This also correlated with a reduced expression of several viral restriction factors, including *SAMHD1*, *HERC5*, *ISG15*, *RSAD1,* and *APOBEC3*, in CHIKV-infected cells lacking IRGM compared to the control cells (Fig 5E–J). Further, the CHIKV-induced levels of ISG15 and ISGylation of other proteins were attenuated in IRGM knockdown cells (Fig 5K). Similarly, the levels of viperin, IFITM3, SAMHD1, MX1, and OAS1 were also reduced in CHIKV-infected IRGM knockdown cells than in the control, in two different cell lines (THP-1 and HT-29) (Fig 5K and L). Furthermore, similar results were obtained when JEV is used instead of CHIKV (Fig 5M) suggesting that IFN response is blunted in viral-infected IRGM knockdown cells.

The decreased levels of ISGs in viral-infected IRGM knockdown cells (or knockout mice) are probably the consequence of a reduced viral uptake and/or propagation in these cells, which results in lower amounts of PAMPs and therefore a blunted host IFN response. To test this hypothesis, we transfected heat-killed CHIKV and viral RNA isolated from CHIKV in control and IRGM knockdown cells before examining ISGs levels by Western blot. Unlike the case of live viruses, the levels of ISGs were not decreased in IRGM-depleted cells (except in the case of heat-killed viruses and viperin)

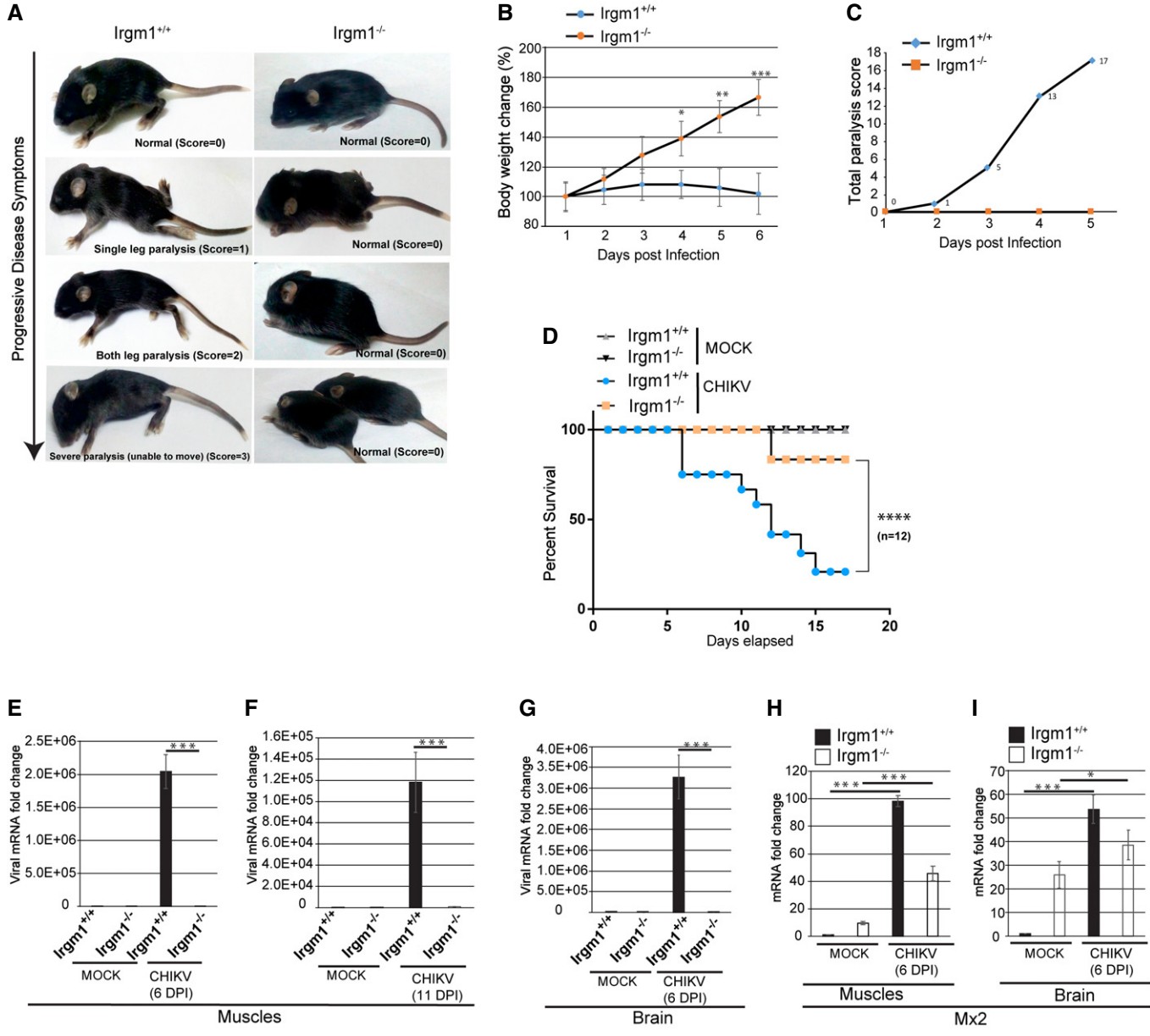

**Figure 4.** *Irgm1⁻ᐟ⁻* mice are resistant to CHIKV infection.

A   The stages of paralysis in C57BL/6 mice post CHIKV infection (MOI 1 × 10⁷ PFU/mouse, 6 dpi) with scoring according to the severity of the disease.

B   The graph depicts percentage change in body weight of CHIKV-infected *irgm1⁺ᐟ⁺* and *irgm1⁻ᐟ⁻* neonates for duration as indicated (*n* = 12, mean ± SD, \*P < 0.05, \*\*P < 0.005, \*\*\*P < 0.0005, Student's unpaired *t*-test).

C   The graph depicts total paralysis scores of CHIKV-infected *irgm1⁺ᐟ⁺* and *irgm1⁻ᐟ⁻* neonates (*n* = 12) until 5 days post-infection.

D   Kaplan–Meier survival graph depicts percentage survival of mock and CHIKV-infected *irgm1⁺ᐟ⁺* and *irgm1⁻ᐟ⁻* neonates during the course of infection (*n* = 12, \*\*\*\*P < 0.0001, Log-rank (Mantel-Cox) test).

E, F   Total RNA isolated from muscles of mock and CHIKV-infected (MOI 1 × 10⁷ PFU/mouse, 6 and 11 dpi) *irgm1⁺ᐟ⁺* and *irgm1⁻ᐟ⁻* mice and was subjected to qRT–PCR to quantitate the viral load (*n* = 3, mean ± SD, \*\*\*P < 0.0005, Student's unpaired *t*-test).

G   Total RNA isolated from the brain of mock and CHIKV-infected (MOI 1 × 10⁷ PFU/mouse, 6 dpi) *irgm1⁺ᐟ⁺* and *irgm1⁻ᐟ⁻* mice and was subjected to qRT–PCR for quantitation of viral load (*n* = 3, mean ± SD, \*\*\*P < 0.0005, Student's unpaired *t*-test). The total RNA used for qRT–PCR with the brain is four times more than muscles.

H, I   Total RNA isolated from muscles and brains of mock and CHIKV-infected (MOI 1 × 10⁷ PFU/mouse) (6 dpi) *irgm1⁺ᐟ⁺* and *irgm1⁻ᐟ⁻* mice and was subjected to qRT–PCR with MX2 (*n* = 3, mean ± SD, \*P < 0.05, \*\*\*P < 0.0005, Student's unpaired *t*-test). The total RNA used for qRT–PCR with the brain is four times more than muscles.

(Fig 5N). The levels of ISG15 and viperin are rather increased in IRGM-depleted cells upon treatment with CHIKV viral RNA (Fig 5N). These data suggest that to some extent our hypothesis is correct; however, it appears that some additional factors contribute to such powerful reduction of ISG's in viral-infected IRGM-depleted cells.

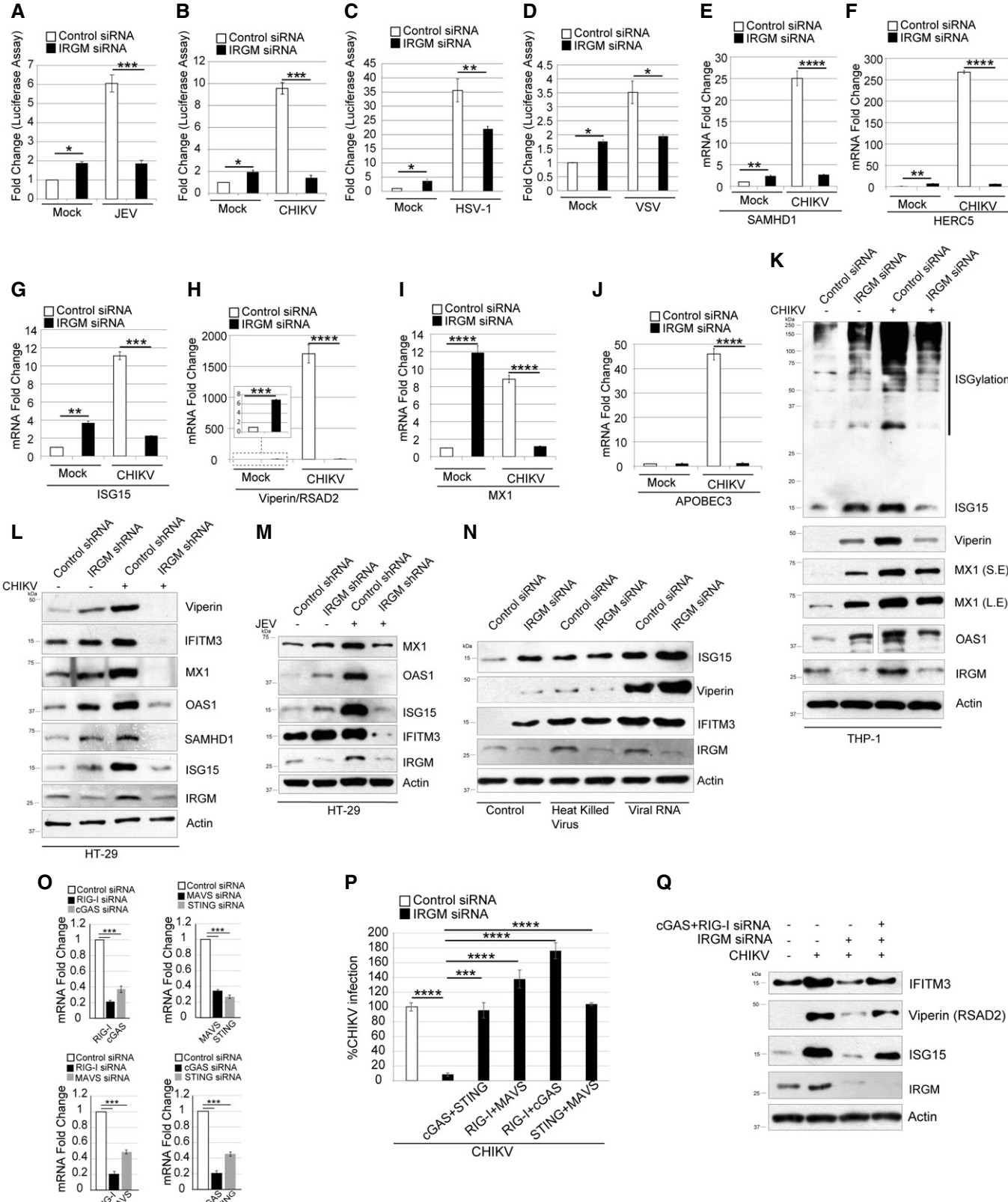

**Figure 5.**

Figure 5. Type I IFN response is blunted in virus-infected IRGM knockdown cells.

A–D  Control and si-IRGM transfected THP-1 IFN reporter cells were kept uninfected (Mock) or infected with (A) JEV (MOI 5) or (B) CHIKV (MOI 5) or (C) HSV-1 (MOI 2.5) or (D) VSV-eGFP (MOI 1) and the supernatant collected 8 hpi were subjected to luciferase assay. The graphs depict fold change in interferon response. (*n* = 3, mean ± SD, *P* < 0.05, **P* < 0.005, ***P* < 0.0005, Student's unpaired *t*-test).

E–J  Control and si-IRGM transfected HT-29 cells uninfected or infected with CHIKV and qRT–PCR analysis were performed with several ISG's (E) SAMHD1 (F) HERC5 (G) ISG15 (H) viperin/RSAD2 (I) MX1 (J) APOBEC3G. (*n* = 3, mean ± SE, **P* < 0.005, ***P* < 0.0005, ****P* < 0.00005, Student's unpaired *t*-test).

K  Western blot analysis with cell lysates of mock and CHIKV (MOI 5, 24 h) infected control and si-IRGM transfected THP-1 cells and probed with the indicated antibodies. S.E, short exposure; L.E, long exposure.

L  Western blot analysis with cell lysates of mock and CHIKV (MOI 5, 24 h) infected control and IRGM knockdown HT-29 cells and probed with the indicated antibodies.

M  Western blot analysis with cell lysates of mock and JEV (MOI 5, 24 h) infected control and si-IRGM transfected HT-29 cells and probed with the indicated antibodies.

N  Western blot analysis with cell lysates of THP-1 control or IRGM knockdown cells, untransfected or transfected with, heat-killed whole CHIKV or CHIKV viral RNA and probed with the indicated antibodies.

O  qRT–PCR analysis to determine the knockdown efficiencies of PRR's and adaptor proteins as indicated (*n* = 3, mean ± SE, ***P* < 0.0005, Student's unpaired *t*-test).

P  qRT–PCR analysis with total RNA isolated from control and IRGM knockdown HT-29 cells transfected with siRNA combinations as indicated that were infected with CHIKV (MOI 5, 24 h). (*n* = 3, mean ± SE, ***P* < 0.0005, ****P* < 0.00005, Student's unpaired *t*-test).

Q  Western blot analysis with cell lysates of mock and CHIKV (MOI 5, 24 h) infected control and IRGM knockdown HT-29 cells transfected with indicated siRNA and probed with the indicated antibodies.

Source data are available online for this figure.

Overwhelming inflammation (cytokine storm) is one of the main reasons for mortality in viral infections, including the one of SARS-CoV-2. These data suggest that IRGM could be an ideal and broad therapeutic target for prophylactic use against a variety of viruses. Importantly, IRGM inhibitors may inhibit viruses without causing an elevated inflammation.

### RIG-I-MAVS and cGAS-STING signaling are required to maintain an antiviral state of IRGM knockdown cells

We have previously shown that both cGAS-STING and RIG-I-MAVS signaling are important for constitutive upregulation of type I IFN response in IRGM-depleted cells (Jena *et al*, 2020). Therefore, we tested whether cGAS-STING and RIG-I-MAVS pathways play a significant role in the antiviral response of IRGM-depleted cells. We performed a simultaneous knockdown of cGAS and STING, RIG-I and MAVS, STING and MAVS or cGAS and RIG-I, in IRGM-depleted THP-1 cells, before infecting them with CHIKV (Fig 5O). The depletion of these PRRs and adaptors in IRGM knockdown cells make them susceptible to viral infection (Fig 5P). Interestingly, the low levels of ISGs in infected IRGM knockdown cells were rescued upon knockdown of cGAS and RIG-I (Fig 5Q). These data suggest that induction of cGAS-STING and RIG-I-MAVS signaling pathways are responsible for the antiviral state of IRGM-deficient cells.

Previously, we found that mitophagy defects in IRGM-depleted cells result in increased reactive oxygen species (ROS), and quenching it with N-acetyl-cysteine (NAC) suppresses IFN response in these cells, but not completely (Jena *et al*, 2020). We tested whether NAC can restore the replication of viruses in IRGM knockdown cells. There was significant but not complete restoration of viral replication in these cells (Fig EV3K), indicating that just ROS production is not enough to increase IFN response and decrease viral replication.

Taken together, this study shows that inhibiting IRGM establishes a broad and strong antiviral immune state in cells. Several mechanisms might have contributed to such a powerful antiviral response, which includes the enhanced expression of almost all viral restriction factors, the presence of upregulated MHC-1 antigen presentation/processing pathway, and the persistent presence of stress granules. It is an unprecedented observation that so many of the key viral restriction factors can be induced by the manipulation of a single gene. This could be because IRGM controls several of the DNA/RNA viral sensing-signaling pathways, including the cGAS-STING, the RIG-I-MAVS, the TLR3-TRIF, and the TLR7 signaling system (Jena *et al*, 2020; Rai *et al*, 2021). IRGM deficiency causes a potent induction of these signaling systems leading to a strong type I IFN-mediated antiviral response.

Previously, it was shown that IRGM could interact with several viral proteins and can modulate replication of Hepatitis C virus (HCV), human immunodeficiency virus 1 (HIV-1), and Measles virus (MeV) (Gregoire *et al*, 2011). The authors suggest that autophagy defects in IRGM-depleted cells could be the reason for lower viral replication in these cells. Our data show that autophagy defects leading to a heightened IFN response resulting in the upregulation of large numbers of host antiviral processes is the prime mechanism of controlling viral replication in IRGM-depleted cells. Altogether, our work along with this previous study strongly suggests that therapeutic targeting of IRGM could lead to the development of host-directed therapies for improving broad antiviral innate immunity against emerging strains and viruses. We also understand that on the flipside, therapeutic inhibition of IRGM may predispose to autoinflammatory conditions. However, it can be argued that since such drugs would be administrated for prophylactic use for a short duration especially during viral outbreaks, the benefits may outweigh the side effects (if there are any).

# Materials and Methods

### Cell culture

The cell lines, HT-29 (ATCC #HTB-38), THP-1 (ATCC #TIB-202), HeLa (ATCC #CCL-2), were obtained from American Type Culture Collection (ATCC). THP-1 dual cells (NF-κB-SEAP and IRF-Lucia luciferase Reporter Monocytes; #thpd-nfis) were purchased from InvivoGen. Huh7 cells were kindly provided by Dr. Tonya Colpitts.

HT-29 and Vero cells were grown in DMEM (Gibco #10569044) supplemented with 10% fetal bovine serum (FBS) and penicillin/ streptomycin (10,000 units/ml). Human monocytic cell line, THP-1, and BMDMs were grown in RPMI-1640 (Gibco#61870127) media supplemented with 10% FBS, 5 mM L-glutamine, glucose (5%), HEPES buffer, sodium pyruvate (1 mM), penicillin/strepto-mycin (10,000 units/ml). All the experiments were performed with cells before the 20[th] passage was reached. The IRGM CRISPR knockout and shRNA stable in cell lines are described before (Jena *et al*, 2020).

## Virus

CHIKV (accession no. EF210157.2) was gifted by Dr. M. M. Parida, DRDE Gwalior, India. JEV (accession no. AF075723) was kindly provided by Dr. A. Basu, National Brain Research Center, India. For ZIKV, we used a clinical isolate from Surinam (a kind gift from Dr. Martijn van Hemert, Leiden University Medical Center). WNV strain NY99 is a kind gift from Dr. Claire Huang, CDC. HSV-1 (accession no. JQ673480.1) was kindly gifted by Dr. Roger Everett, Glasgow University. The SARS-CoV-2 strain (accession no. EPI_ISL_1196305) was isolated and characterized previously (Raghav *et al*, 2020).

## Virus infection

Control and IRGM knockdown cells were seeded in 6-well plates. Next day, the cells were infected with either CHIKV, JEV, or HSV-1 with MOI as indicated in figure legends and as described earlier (Kumar *et al*, 2014; Nayak *et al*, 2017). Briefly, the confluent mono-layers were washed in sterile 1X PBS and infected with the virus (di-luted in serum-free media) for 1.5 h with manual shaking at an interval of 10 min. Then, the cells were washed in sterile 1X PBS and were maintained in complete DMEM in the incubator till harvest. The infected cells were collected for Western blotting or qRT–PCR analysis. THP-1 or THP-1 dual control and IRGM knock-down cells were differentiated into a macrophage-like state by the addition of 50 ng/ml of PMA (16 h). After a resting period of 48 h, macrophages were infected with viruses.

## Plaque assay

The plaque assay was performed to quantitate the release of new infectious viral particles according to the previously described proto-col (Kumar *et al*, 2014). Briefly, virus-infected cell culture super-natants were serially diluted in serum-free media, which was used to infect confluent Vero cells seeded in 12-well cell culture plates. After infection, the cells were overlaid with methylcellulose (Sigma #M0387) containing DMEM and kept in the 37°C incubator for 3–4 days until visible plaques developed. Then the cells were fixed in 8% formaldehyde, washed gently in tap water followed by staining with crystal violet. The number of plaques was counted manually to determine plaque-forming units/ml (PFU/ml).

## Transient transfection with siRNA

The THP-1 cells were electroporated (Neon, Invitrogen #MPK5000; setting: 1400 V, 10 ms, 3 pulses using 100 µl tip #MPK10096) with non-targeting siRNA (30 nM) or specific siRNA (30 nM) and

incubated for 24 h. Another round of siRNA transfection was performed after 24 h in a similar condition and incubated for the next 48 h. The siRNA (10 nM) transfection in HT-29 cells was performed using the Lipofectamine® RNAiMAX (Invitrogen #13778075) as per the manufacturer's instructions. Following siRNA was used in the present study: Non-specific siRNA (SMARTpool: siGENOME ns siRNA; Dharmacon #D-001206-13-20), IRGM siRNA (SASI_HS02_00518571), Human RIG-I siRNA (SASI_Hs01_00047980), Human cGAS siRNA (SASI_Hs01_00197466), MAVS siRNA (SASI_Hs01_00128708), STING siRNA (SASI_Hs02_00371843).

## Transient transfection with plasmids

For transient expression, THP-1 cells were transfected using the Neon electroporation system (Thermo) with the following parame-ters: 1,300 V, 30 ms, one pulse. For rescue experiments, $2 \times 10^6$ THP-1 cells were transfected with IRGM (30 nM) siRNA. After 72 h, the cells were transfected with 3X-Flag or Flag IRGM (3 µg). After 4 h, cells were collected for qRT–PCR or Western blotting. For the catalytic mutant overexpression assays in THP-1, $2 \times 10^6$ THP-1 cells were transfected with 3X-Flag or Flag IRGM (3 µg) or FLAG IRGM S47N (3 µg). After 4 h, cells were collected for qRT–PCR. Plasmid transfection in HT-29 cells was performed using ViaFect reagent (Promega #E4982) as per the manufacturer's instruction.

## Western blotting

The cell lysates were prepared using the NP-40 lysis buffer (Invitro-gen #FNN0021) containing protease inhibitor cocktail (Roche #11836170001), phosSTOP (Roche #4906845001), and 1 mM PMSF (Sigma #P7626). Mice tissues (100 mg) were homogenized in 1 ml Radio-immunoprecipitation assay (RIPA) buffer (20 mM Tris, pH 8.0; 1 mM, EDTA; 0.5 mM, EGTA; 0.1% Sodium deoxycholate; 150 mM NaCl; 1% IGEPAL; 10% glycerol) with protease inhibitor cocktail, phosSTOP and 1 mM PMSF using tissue tearor (BioSpec #985370). Lysates were separated using SDS-polyacrylamide gel, transferred onto a nitrocellulose membrane (Bio-Rad), and blocked for 1 h in 5% skimmed milk followed by incubation in primary anti-body overnight at 4°C. Membranes were then washed thrice with 1X PBS/PBST and incubated for 1 h with HRP-conjugated secondary antibody. After washing with PBS/PBST, the blots were developed using enhanced chemiluminescence reagent (Thermo Fisher #32132X3).

## Antibodies

Primary antibodies used in Western blotting with dilutions: Actin (Abcam #ab6276; 1:5,000), IRGM antibody rodent specific (CST #14979; 1:1,000), IRGM (Abcam #ab69494; 1:500), MX1 (CST #37849; 1:1,000), OAS1 (CST #14498; 1:1,000), ISG15 (CST #2743; Santacruz sc-166755; 1:1,000; 1:1,000), SAMHD1 (CST #12361; 1:1,000), BST2 (CST #19277; 1:1,000), viperin (CST #13996; 1:1,000), APOBEC3G (CST #43584; 1:1,000), IFITM3 (CST #59212; 1:1,000), PKR B10 (sc-6282; 1:1,000), p-PKR (#ab32036; 1:1,000), EIF2 Alpha (CST #9722; 1:1,000), p-EIF2 Alpha (CST #3398S; 1:1,000), G3BP (BD Bioscience #611126; 1:1,000), GAPDH (CST #2118; 1:1,000), ACE2 (R&D systems #AF933; 1:1,000), HRP-conjugated secondary antibodies were purchased from Santa Cruz

(1:2,000) or Promega (1:5,000) or Abcam (1:10,000) or Novus (1:5,000).

Primary antibodies used in immunofluorescence assays with dilutions: dsRNA (Kerafast # ES2001; 1:60), TIA-1 (Abcam #140595; 1:100), OVA257-264 (SIINFEKL) peptide bound to H-2Kb Monoclonal Antibody (eBio25-D1.16 25-D1.16), PE (Invitrogen).

## Immunofluorescence analysis

Approximately, $10^5$ cells were seeded on the coverslip and allowed to adhere to the surface. For THP-1, cells were differentiated into the macrophage-like state by the addition of 50 ng/ml of phorbol 12-myristate 13-acetate (PMA) (Sigma #P8139) for 16 h. Next, the culture medium was replaced and incubated for 48 h. The adhered cells were fixed in 4% paraformaldehyde for 10 min and permeabilized with 0.1% Triton X-100 for 10 min, followed by blocking with 1% BSA for 30 min at RT. The cells were then incubated with primary antibody for 1 h at RT, washed thrice with 1X PBS, followed by 1 h incubation with Alexa Fluor-conjugated secondary antibody. Cells were again washed thrice with 1X PBS, mounted (Prolong gold antifade, Invitrogen #P36931), and visualized using Leica TCS SP8 STED confocal microscope. For immunofluorescence analysis, ovalbumin pulsed BMDMs were once washed with 1X PBS and stained with PE (Phycoerythrin)-conjugated 25-D1.16 monoclonal antibody (0.2 µg/test) for 2 h at 4°C and subsequently used for microscopy.

## Enzyme-linked immunosorbent assay (ELISA)

ELISA was performed where polystyrene 96-well plates (Corning Costar # 9018) were pre-coated overnight at 4°C with mouse serum (antigen) diluted in coating buffer i.e. 1X PBS, then blocked with 3% BSA diluted in 1X PBS for 2 h at RT, followed by incubation with primary antibody IFN-β antibody 7F-D3 (sc-57201; 1:50) for 2 h at RT. Primary antibody was removed and washed with 1X PBS containing 0.05% Tween 20 followed by incubation with HRP-conjugated secondary antibody for 1 h at RT and washed again. 1X TMB (3,3′,5,5′-Tetramethylbenzidine) was added, and the plate was incubated in the dark, and the reaction was stopped using 1 M $H_3PO_4$. Microplate reader (Bio-Rad) was used to measure the absorbance at 450 nm.

## Luciferase assay

Luciferase assay was performed using THP-1 dual cells (thpd-nfis, InvivoGen) as per manufacture protocol (InvivoGen). Briefly, THP-1 dual control and IRGM knockdown cells were differentiated into a macrophage-like state by the addition of 50 ng/ml of PMA (16 h). After a resting period of 48 h, macrophages were infected with viruses or treated with Poly I:C HMW (1 µg/ml), Poly dA:dT (1 µg/ml), cGAMP (1 µg/ml), and IFN-β (500 ng/ml) for indicated time points and the supernatant were collected. All PAMPs were purchased from InvivoGen. 10–20 µl of the sample was pipetted per well into a 96-well white plate. 50 µl of QUANTI-Luc™ luminescence assay reagent; # rep-qlc2, (InvivoGen) was added to each well. The plate was gently tapped to mix and immediately proceeded with the measurement of luminescence using the PerkinElmer VICTOR Nivo multimode plate reader.

## RNA isolation and quantitative real-time PCR

RNA was extracted from cells using TRIZOL by following the manufacturer's protocol. Viral RNA from the supernatant was extracted by an automated RNA isolation system. The cDNA was synthesized using the high capacity DNA reverse transcription kit (Applied Biosystems,#4368813), and qRT–PCR was performed using TaqMan master mix (Applied Biosystems, #4369016) or Power SYBR green PCR master mix (Applied Biosystems, #4367659) according to the manufacturer's protocol. For normalization of the assay, the housekeeping gene GAPDH or β-Actin was used. The fold change in expression was calculated by the $2^{-\Delta\Delta Ct}$ method. Excel or GraphPad is used for generating graphs and statistics.

## Mouse experiments

*Irgm1* knockout (C57BL/6) mice ($Irgm1^{-/-}$) were kindly provided by Dr. Gregory Taylor and maintained as described previously (Liu *et al*, 2013). Mouse experiments were performed with procedures approved by the institutional animal ethical committee (IAEC) at the Institute of Life Sciences, Bhubaneswar, India. For each experiment, littermates were used with no gender bias. The six days old *Irgm1* wild-type and knockout mouse pups ($n = 12$) were infected with CHIKV ($1 \times 10^7$ PFU/mouse) through the intradermal route. Mice were monitored daily for muscle weakness, flaccid paralysis symptoms, and changes in body weights. The paralysis scores were assigned as follows: normal; 0, single-leg paralysis; 1, both leg paralysis; 2, severe paralysis; 3. The mice were sacrificed, and different organs were collected and processed further for qRT–PCR analysis.

## Mice bone marrow cells isolation and differentiation into macrophages

The bone marrow cells from wild-type ($Irgm1^{+/+}$) and knockout ($Irgm1^{-/-}$) mice were isolated and differentiated into macrophages by standard procedure. Briefly, six to eight weeks old male C57BL/6 $Irgm1^{+/+}$ and $Irgm1^{-/-}$ mice were sacrificed by cervical dislocation, bone marrow cells from the tibia, and femurs were flushed out in RPMI medium. Red blood cells were removed by cell lysis buffer containing (155 mM $NH_4Cl$, 12 mM $NaHCO_3$, and 0.1 mM EDTA). Bone marrow cells were differentiated in RPMI medium (10% FBS, 1 mM sodium pyruvate and 0.05 M 2-mercaptoethanol) containing 20 ng/ml mouse M-CSF (Gibco #PMC2044) for 5 days. On every alternate day, media was replaced with fresh media containing M-CSF.

## Antigen uptake assay

The cells were incubated with 5 µg/ml fluorescent Alexa Fluor 488-OVA (a model antigen) (Invitrogen #O34781) or 10 µg/ml Alexa Fluor™ 488-conjugated transferrin (Invitrogen #T13342) for 30 min at 37°C. The cells were serum-starved for 1 h in case of treatment with transferrin. Any excess fluorochrome bound to the cell surface was quenched for 3–4 min on ice using 2% FBS/ PBS solution. After repeating quenching steps twice, cells were thoroughly washed using ice-cold FACS buffer (2% FBS/ PBS) and then immediately examined using a FACS (BD LSRFortessa). Data

were analyzed using the FlowJo Software (Tree Star Inc., Ashland, OR).

**Antigen processing/presentation assay**

The cells were treated with BODIPY-conjugated DQ-OVA (# D12053 Invitrogen), a self-quenched conjugate of OVA that exhibits bright green fluorescence only upon proteolytic cleavage releasing the dye molecule from the OVA. Briefly, the cells were incubated with 10 μg/ml of DQ-OVA for 30 min at 37°C. After 30 min, cells were washed using 1X PBS and then fixed with 2% paraformaldehyde for 10 min at room temperature. Fixed cells were washed twice using ice-cold FACS buffer and then analyzed using a FACS (BD LSRFortessa). Negative untreated control cells were separately prepared by incubation of cells with DQ-OVA on ice followed by fixation. The MFI of the ice control cells was subtracted from that of cells incubated at 37°C with OVA per treatment or control. Data were analyzed using the FlowJo Software (Tree Star). For immunofluorescence, cells were subjected to 10 μg/ml DQ-OVA treatment for 30 min at 37°C followed by washing two times with 1X PBS and then fixed with 2% paraformaldehyde for 10 min at room temperature. The coverslips were washed thrice with 1X PBS and permanently mounted using Prolong gold antifade (Invitrogen #P36931) and finally visualized using TCS SP8 STED confocal microscope.

**SIINFEKL-peptide-based system to measure MHC Class I antigen presentation**

The 25-D1.16 monoclonal antibody reacts with the OVA-derived peptide SIINFEKL bound to H-2Kb of MHC class I, but not with unbound H-2Kb, or H-2Kb bound with an irrelevant peptide. The BMDMs were treated with OVA (2 mg/ml) for 3 h at 37°C. Cells were then washed once with 1X PBS and stained with PE (Phycoerythrin)-conjugated 25-D1.16 monoclonal antibody (0.2 μg/test) for 30 min at 4°C followed by FACS analysis. For immunofluorescence analysis, OVA pulsed BMDMs were once washed with 1X PBS and stained with PE (Phycoerythrin)-conjugated 25-D1.16 monoclonal antibody (0.2 μg/test) for 2 h at 4°C and subsequently used for microscopy.

**Software and statistics analysis**

Microsoft Excel and GraphPad Prism 6 is used to analyze and present the data. The statistical test used is mentioned in respective figure legends. For making graphics Biorender.com is used.

# Data availability

RNA sequencing, data processing, and gene expression analysis were described previously (Jena *et al*, 2020). In the current study, we have reanalyzed the previous data (Jena *et al*, 2020) to extract the expression profile of viral restriction factors. Heatmap was generated as described earlier (Jena *et al*, 2020). For comparison with SARS-CoV-2 studies, the genes that are induced significantly (*P* < 0.05; 1.5 folds) in IRGM-depleted cell (Jena *et al*, 2020) are used. The RNA-seq datasets have been deposited in the ArrayExpress database at EMBL-EBI (www.ebi.ac.uk/arrayex press) under accession number E-MTAB-9164 (http://www.ebi.ac.uk/arrayexpress/experiments/E-MTAB-9164) and E-MTAB-9142(http://www.ebi.ac.uk/arrayexpress/experiments/E-MTAB-9142).

**Expanded View** for this article is available online.

## Acknowledgements

This work is funded by the Department of Science & Technology SERB grant CRG/2020/003480, Department of Biotechnology grant BT/PR23942/BRB/10/1808/2019, and DBT/Wellcome Trust India Alliance (IA/I/15/2/502071) fellowship to Santosh Chauhan. Soma Chattopadhyay is funded by the Department of Science and Technology (DST), New Delhi, India, vide-grant no EMR/2016/000854. Fulvio Reggiori is supported by ENW KLEIN-1 (OCENW.KLEIN.118), ZonMW TOP (91217002) Marie Skłodowska-Curie Cofund (713660) and Marie Skłodowska-Curie ETN (765912) grants. Subhash Mehto is supported by the DST-Inspire faculty fellowship (DST/INSPIRE/04/2019/001857). We acknowledge the technical assistance of Bhabani Sahoo (Microscopy facility) and Paritosh Nath (FACS facility). We gratefully acknowledge the support of the Institute of Life Sciences central facilities funded by the Department of Biotechnology (India). We gratefully acknowledge Dr. M. M. Parida, Dr. A. Basu, Dr. Roger Everett for providing CHIKV strain, JEV strain HSV-1 KOS strain, respectively.

## Author contributions

SChau secured funding, conceived the project, designed the experiments, and wrote the manuscript. P.N., N.R.C, and K.K.J performed the majority of experiments. A.D, N.D.K, R.B, KCM, S.M, S.D, T.K.N, S.P, K.R, R.B, performed the experiments. M.K, P.P, S.P, F.R, SChat provided critical inputs for experiments and edited the manuscript.

## Conflict of interest

The authors declare that they have no conflict of interest.

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
