## [Review Process File · EMBO Reports]

Inhibition of IRGM Establishes a Robust Antiviral Immune State to Restrict Pathogenic Viruses

Parej Nath, Nishant Chauhan, Kautilya Jena, Ankita Datey, Nilima Kumar, Subhash Mehto, Saikat De, Tapas Nayak, Swatishmita Priyadarsini, Kshitish Rout, Ramyasingh Bal, Krushna Murmu, Manjula Kalia, Srinivas Patnaik, Punit Prasad, Fulvio Reggiori, Soma Chattopadhyay, and Santosh Chauhan
DOI: 10.15252/embr.202152948

Corresponding author(s): Santosh Chauhan (schauhan@ils.res.in) , Soma Chattopadhyay (soma@ils.res.in)

Review Timeline:

Submission Date:	27th Mar 21
Editorial Decision:	30th Apr 21
Revision Received:	10th Jul 21
Editorial Decision:	9th Aug 21
Revision Received:	13th Aug 21
Accepted:	16th Aug 21

Editor: Achim Breiling

Transaction Report:

Dear Dr. Chauhan,

Thank you for the submission of your research manuscript to EMBO reports. We have now received the reports from the referees that were asked to evaluate your study, which can be found at the end of this email.

As you will see, the referees think that these findings are of interest. However, they have several comments, concerns and suggestions, indicating that a major revision of the manuscript is necessary to allow publication of the study in EMBO reports. As the reports are below, and all their points need to be addressed, I will not detail them here.

Given the constructive referee comments, we would like to invite you to revise your manuscript with the understanding that all referee concerns must be addressed in the revised manuscript or in the detailed point-by-point response. Acceptance of your manuscript will depend on a positive outcome of a second round of review. It is EMBO reports policy to allow a single round of revision only and acceptance of the manuscript will therefore depend on the completeness of your responses included in the next, final version of the manuscript.

Revised manuscripts should be submitted within three months of a request for revision. We are aware that many laboratories cannot function at full efficiency during the current COVID-19/SARS-CoV-2 pandemic and we have therefore extended our 'scooping protection policy' to cover the period required for full revision. Please contact me to discuss the revision should you need additional time, and also if you see a paper with related content published elsewhere.

- 1) a .docx formatted version of the final manuscript text (including legends for main figures, EV figures and tables), but without the figures included. Please make sure that changes are highlighted to be clearly visible. Figure legends should be compiled at the end of the manuscript text.
- 2) individual production quality figure files as .eps, .tif, .jpg (one file per figure), of main figures and EV figures. Please upload these as separate, individual files upon re-submission.

The Expanded View format, which will be displayed in the main HTML of the paper in a collapsible format, has replaced the Supplementary information. You can submit up to 5 images as Expanded View. Please follow the nomenclature Figure EV1, Figure EV2 etc. The figure legend for these should be included in the main manuscript document file in a section called Expanded View Figure Legends after the main Figure Legends section. Additional Supplementary material should be supplied as a single pdf file labeled Appendix. The Appendix should have page numbers and needs

to include a table of content on the first page (with page numbers) and legends for all content. Please follow the nomenclature Appendix Figure Sx, Appendix Table Sx etc. throughout the text, and also label the figures and tables according to this nomenclature.

See also our guide for figure preparation:

http://wol-prod-cdn.literatumonline.com/pb-assets/embosite/EMBOPress_Figure_Guidelines_061115-1561436025777.pdf

4) a complete author checklist, which you can download from our author guidelines (<https://www.embopress.org/page/journal/14693178/authorguide>). Please insert page numbers in the checklist to indicate where the requested information can be found in the manuscript. The completed author checklist will also be part of the RPF.

Please also follow our guidelines for the use of living organisms, and the respective reporting guidelines: <http://www.embopress.org/page/journal/14693178/authorguide#livingorganisms>

5) that primary datasets produced in this study (e.g. RNA-seq, ChIP-seq, structural and array data) are deposited in an appropriate public database. If no primary datasets have been deposited, please also state this a dedicated section (e.g. 'No primary datasets have been generated and deposited'), see below.

The accession numbers and database should be listed in a formal "Data Availability " section (placed after Materials & Methods) that follows the model below. This is now mandatory (like the COI statement). Please note that the Data Availability Section is restricted to new primary data that are part of this study.

Data availability

- RNA-Seq data: Gene Expression Omnibus GSE46843

(<https://www.ncbi.nlm.nih.gov/geo/query/acc.cgi?acc=GSE46843>)

- [data type]: [name of the resource] [accession number/identifier/doi] ([URL or identifiers.org/DATABASE:ACCESSION])

6) We strongly encourage the publication of original source data with the aim of making primary data more accessible and transparent to the reader. The source data will be published in a separate source data file online along with the accepted manuscript and will be linked to the relevant figure. If you would like to use this opportunity, please submit the source data (for example scans of entire gels or blots, data points of graphs in an excel sheet, additional images, etc.) of your key experiments together with the revised manuscript. If you want to provide source data, please include size markers for scans of entire gels, label the scans with figure and panel number, and send one PDF file per figure.

8) Regarding data quantification and statistics, please make sure that, where applicable, the number "n" for how many independent experiments were performed and the type of replicate (biological or technical), the bars and error bars (e.g. SEM, SD) and the test used to calculate p-values is indicated in the respective figure legends. Please provide statistical testing where applicable, and also add a paragraph detailing this to the methods section. See: <http://www.embopress.org/page/journal/14693178/authorguide#statisticalanalysis>

9) Please note our new reference format:
<http://www.embopress.org/page/journal/14693178/authorguide#referencesformat>

10) For microscopic images, please add scale bars of similar style and thickness to all the microscopic images, using clearly visible black or white bars (depending on the background). Please place these in the lower right corner of the images. Please do not write on or near the bars in the image but define the size in the respective figure legend.

11) Please order the manuscript sections like this:
Title page - Abstract - Introduction - Results/Discussion - Materials and Methods - Data availability section - Acknowledgements - Author contributions - Conflict of interest statement - References - Figure legends - Expanded View Figure legends.

12) Please provide a more comprehensive and shorter title (with not more than 100 characters including spaces).

Finally, please note that all corresponding and co-corresponding authors are required to supply an ORCID ID for their name upon submission of a revised manuscript. Please find instructions on how to link the ORCID ID to the account in our manuscript tracking system in our Author guidelines: <http://www.embopress.org/page/journal/14693178/authorguide#authorshipguidelines>

I look forward to seeing a revised version of your manuscript when it is ready. Please let me know if you have questions or comments regarding the revision.

Yours sincerely,

Achim Breiling
Editor
EMBO Reports

Referee #1:

In their manuscript entitled „Inhibiting IRGM Expression Establishes a Robust Antiviral Immune State that can Restrict Infection of Several Pathogenic Viruses" Chauhan and co-workers analysed the function of IRGM in anti-viral responses. They show that depletion of IRGM induces interferon stimulated gene (ISG) expression in human and murine cells. Most importantly they provide several lines of evidence to show that this results in restriction of viral replication of several clinically relevant RNA and DNA viruses, including HSV-1, CHIKV and SARS-CoV-2 both in vitro and in vivo.

The immunity related GTPase M (IRGM) recently gained much attention due to genetic link to autoimmune and autoinflammatory disorders. IRGM has multiple cellular functions, best known its contribution to autophagy. Another group and the authors recently showed that IRGM deficiency leads to a mitochondrial DNA-dependent induction of type I interferon responses (Rai et al. 2021, Jena et al. 2020). Here the authors expand on these findings and examined the effect of IRGM on viral induced interferon responses and viral restriction in cells and in vivo.

The work provides a striking phenotype of the IRGM knock-down and knockout on ISG expression, antigen presentation and viral restriction. This suggests that IRGM might be a useful target for novel anti-viral host-directed therapies. The presented data and the analysis of IRGM in viral infection are novel and relevant and of interest to a broad readership.

The presented data are of high quality and the manuscript is very well written. This reviewer only has a few minor issues that need to be addressed.

Major points:

1. The main criticism of this reviewer is that the authors used different cell types, but it is not clear for the reader why these were used. Could the authors provide the reader some more information and label the cells in the figures?

In Fig. 1 the infection experiments were conducted in colon HT29 cells but all PAMP stimulation was performed only in myeloid THP1 cells. Could the authors perform at least some viral infection also with THP1 cells and compare the PAMP stimulation kinetics shown in panels G-I?

2. The effect of CHIKV infection on ISG levels shown in Fig.5 is extremely high. For this reviewer, the interpretation of these data however is not trivial as the authors used two different cell lines (HT-29 shRNA and siRNA treatment of HT-29 cells). For the data shown in Fig. 5O it would be informative to see also the interferon release and the effect of cGAS/RIG-I/MAVS/STING ko in control siRNA treated cells.

It might be possible that this effect is a particularity of CHIKV. Could the authors address this and repeat the experiment shown in Fig.5L with one of their other viruses?

3. The results shown for APOBEC3 in Fig.2B and Fig.5J are highly different. Is this related to

differences in the responsiveness of the shRNA HT-29 cells and the siRNA treatment? Again, this adds to the confusion of the differed cellular models used. The authors need at least to comment on this and explain why different cellular models were used.

4. The work relies on the use of reporter cell lines and mRNA analysis to measure IFN induction. Could the authors could measure secreted type I interferon in selected critical assays and for the in vivo work (Fig. 4H,I; Fig. 5 A-F, Fig.5O)?

5. As small samples sizes are used individual data points should be plotted in the graphs.

6. Fig. 4D: Kaplan-Meier estimator should be used for statistical analysis and data shown as a Kaplan-Maier plot.

7. Fig. 5 G-I: The authors nicely show that several MHC class I genes are regulated. It was shown that all of these are targets of NLRC5 which is a main regulator of MHC class I genes and an interferon target gene. It is thus likely that NLRC5 expression is the cause of the observed phenotype. Could the authors analyse NLRC5 mRNA expression in their samples or at least comment on this in the text.

8. Interferon responses can affect cell differentiation. As the authors used differentiation protocols for THP1 cells and BMDMs this could be a cofounder. Could the authors show that IRGM/Irgm1 deletion/depletion does not affect macrophage differentiation?

9. Polymorphisms in IRGM and its promotor are linked to autoimmune disease and Crohn's disease and the induction of type I interferon repones by IRGM loss-of function might be a key event here. This should be mentioned as a word of caution when proposing targeting of IRGM a strategy for antiviral therapy.

Minor points:

- Description of the statistical analysis and software is missing in the material methods.
- Lines 132-148: The description of the ISG functions is rather long and in view of this reviewer not of help for the reader. The authors might reduce this part and add more discussion on the cell types used. Along these lines: Fig.2: Personally, this reviewer finds the panels A and F not very helpful, and they take a lot of space.
- Line 224: Remove statement on plaque assay as this is evident.
- Line 229: Typo "evaluated".
- Lines 325-326: The indicated panels of Fig.4 are mixed up.
- Line 337: "...the type I IFN response...."
- Fig. 4C: It would help to show all the different scores instead of a compound score.
- Fig. 5N: Please label the mRNA targets on the graph axes for clarity.

Referee #2:

This manuscript is a follow-up to a 2020 EMBO Reports paper from this group in which they demonstrated that IRGM1 interacts with cellular nucleic acid sensing proteins to direct them to the autophagy pathway to thus limit spontaneous interferon production. Here they confirm many aspects of their prior publication, and show that spontaneous interferon produced when IRGM1 is knocked down inhibits replication of many viruses in vitro. They further show that IRGM1^{-/-} mice are resistant to Chikungunya virus in vivo. In tangentially related experiments, they also show that antigen processing pathways are also increased when IRGM1 is deficient. Conceptually and mechanistically, this paper does not offer much of an advance beyond their previous paper, but it does contain an abundance of experiments and data that are well done and convincing.

1. My only major concern with the data is that this group claims to have infected THP1 cells with SARS-CoV-2 when no other group in the world (to my knowledge) has been able to do so.

Referee #3:

The work by Nath et al. displays that the reduced expression of IRGM in human cells, as well as the complete extinction of the related mouse *Irgm1* protein, confer a very potent antiviral status to human cells/mice, respectively. IRGM deficiency correlates with the overexpression of multiple innate and adaptive immunity-related proteins and functions. As a consequence infection by several RNA and DNA viruses are much better controlled in cells with low IRGM expression. Additionally, *Irgm1* KO neonates are highly resistant to CHIK virus infection, by contrast to their wt counterpart.

The effect of the deficiency of IRGM on immune functions is impressive. As written by the authors "it is an unprecedented observation that so many of the key viral restriction factors can be induced by manipulation of a single gene".

Some points could be done and or clarified

KO/rescue experiments could be performed to display the specificity of the si/shIRGM treatments.

Fig 1 A-F complete western blot should be shown as supplementaries (not just the cutted bands).

IRGM reduction has already been reported to affect replications of several viruses. This work has to be referenced and discussed (PloSPathogens, 2011, 7, 12).

Since IRGM can be detected by western blot, protein expression has to be shown in siIRGM and shIRGM-treated cells (not only mRNA fold changes).

Fig 1L: Type I IFN response is regulated through different pathways by viral PAMPs and by IFN- β . How the authors explained that both signals increased type I IFN response in siIRGM cells ?

For a reader, it is not easy to understand how one single protein can as efficiently impact processes as different than internalisation / degradation / antigen presentation and type I IFN-related antiviral

responses. Does the authors found one immune-related function not altered by the absence of IRGM ?

Since many of the process analysed were reported to be induced by oxidative stress the oxidative status of IRGM defective cells should be looked at.

Since viral replication is affected by the absence of IRGM, we could ask whether the expression of the surface receptors of the tested viruses are as efficiently expressed on IRGM+ and IRGM- cells ? (what could impact virus entry)

The reason of the supplementary fig 3C/D/E is not obvious and unnecessary.

Do IRGM1-/- mice have a higher titer of seric type I IFN than wt mice at steady stae ?

Overexpression of IRGM could be tested for its impact on antiviral gene expression, and virus replication.

Referee #4:

Nath et al. report findings that extend their previous discoveries about the regulation of innate sensing by IRGM. Here, they show that loss of IRGM confers cells with resistance to infection by a diverse set of viruses. They associate this resistance with a high degree of interferon-stimulated gene induction when IRGM is silenced or knocked out. The authors include a huge amount of data in this manuscript, which is appreciated, but it seems some conclusions are premature. Overall, the relevance of their findings to human health remains uncertain because IRGM is an important negative regulator of spontaneous and infection-induced innate immunity, including inflammation.

Major:

Figure 1: The authors show time-course experiments using viral PAMPs indicating that IRGM is induced over time. However, the authors only show the effect of virus infection on IRGM at a single time point. Considering that the timing of IRGM induction is important to our understanding of how cells regulate the induction of antiviral ISGs, this is a crucial experiment.

Figure 5: It is strange that no Viperin mRNA is detected in panel H, while protein is detected in panel L, and that protein level is increased upon IRGM shRNA. Do these data suggest that IRGM can impact not only mRNA levels of ISGs but also protein levels (by affecting ISG protein turnover)?

Line 351: The language used by authors is unclear and so I am unsure of whether the authors believe that virus replication is necessary in order for ISGs to be induced in this experiment. Furthermore, there are differential effects on the ISGs examined, suggesting that a blanket statement is inappropriate here. For example, Viperin protein levels are clearly enhanced by replicating virus and hardly at all by heat killed virus, while IFITM3 protein levels are induced by both replicating virus and heat killed. These data seem to suggest that certain ISGs are induced following replication while others are induced by PAMPs present in heat killed virus.

In this combined Results and Discussion format, the authors don't have the opportunity to discuss the trade-offs of targeting IRGM as a host-directed therapy. For example, how would the authors plan to minimize autoimmune-related disease resulting from unchecked interferon production, in the

absence and presence of virus infection?

The authors do not discuss how the GTPase function of IRGM contributes to its regulation of ISGs.

Figure 2: Can the authors also test how BMDM take up transferrin-AF 488 and LDL-AF 488? That way, the authors could distinguish between an effect of IRGM on endocytic activity versus a selective effect on ovalbumin-AF 488 uptake.

Throughout the figures, IRGM immunoblots are not shown following IRGM silencing or knockout, for some reason.

Minor:

Put lines around western blot boxes for increased readability.

Figure 1T: One of the columns is white when it should be black.

Figure 3: the term "viral load" is inappropriate when referring to the extent of virus infection detected in cell culture. Instead, the authors should use "% Envelope+ cells" and "% Nucleocapsid+ cells" on the Y axes.

Referee #1:

In their manuscript entitled "Inhibiting IRGM Expression Establishes a Robust Antiviral Immune State that can Restrict Infection of Several Pathogenic Viruses" Chauhan and co-workers analysed the function of IRGM in anti-viral responses. They show that depletion of IRGM induces interferon stimulated gene (ISG) expression in human and murine cells. Most importantly they provide several lines of evidence to show that this results in restriction of viral replication of several clinically relevant RNA and DNA viruses, including HSV-1, CHIKV and SARS-CoV-2 both in vitro and in vivo.

The immunity related GTPase M (IRGM) recently gained much attention due to genetic link to autoimmune and autoinflammatory disorders. IRGM has multiple cellular functions, best known its contribution to autophagy. Another group and the authors recently showed that IRGM deficiency leads to a mitochondrial DNA-dependent induction of type I interferon responses (Rai et al. 2021, Jena et al. 2020). Here the authors expand on these findings and examined the effect of IRGM on viral induced interferon responses and viral restriction in cells and in vivo.

The work provides a striking phenotype of the IRGM knock-down and knockout on ISG expression, antigen presentation and viral restriction. This suggests that IRGM might be a useful target for novel anti-viral host-directed therapies. The presented data and the analysis of IRGM in viral infection are novel and relevant and of interest to a broad readership.

The presented data are of high quality and the manuscript is very well written. This reviewer only has a view minor issues that need to be addressed.

Response: We are very thankful to the reviewer for reading our manuscript thoroughly and for the appreciation. We have now further improved our manuscript by carefully addressing the constructive comments of the reviewers and performing a large number of experiments. Thanks a lot for your time and consideration!

Major points:

1. The main criticism of this reviewer is that the authors used different cell types, but it is not clear for the reader why these were used. Could the authors provide the reader some more information and label the cells in the figures?

In Fig. 1 the infection experiments were conducted in colon HT29 cells but all PAMP stimulation was performed only in myeloid THP1 cells. Could the authors perform at least some viral infection also with THP1 cells and compare the PAMP stimulation kinetics shown in panels G-I?

Response: Thanks for the concern. We would like to bring to the kind notice of the reviewer that IRGM is expressed both in myeloid cells and epithelial cells. We used THP-1 (monocytic cell line) and HT-29 (colon epithelial cell line) cells here. In our previous experience (Chauhan et al., 2015, Molecular Cell; Mehto et al., 2019, Molecular Cell; Jena et al., 2020, EMBO reports), we didn't find much difference in immune responses in monocytic cell lines vs epithelial cell lines, whenever IRGM is perturbed. Also, IRGM behaves more or less similarly in these cell lines when stimulated by PAMPs.

Nevertheless, following the reviewer's advice we have now performed the experiments in THP-1 cells and determined the kinetics of IRGM induction after infecting the cells with HSV-1, CHIKV, and JEV. In addition, we have also performed PAMPs stimulation kinetics with 5'pppRNA and poly IC (Figure 1A-1F) in THP-1 cells.

Thanks a lot for the suggestions!

Figure 1A-F

2. The effect of CHIKV infection on ISG levels shown in Fig.5 is extremely high. For this reviewer, the interpretation of these data however is not trivial as the authors used two different cell lines (HT-29 shRNA and siRNA treatment of HT-29 cells). It might be possible that this effect is a particularity of CHIKV. Could the authors address this and repeat the experiment shown in Fig.5L with one of their other viruses?

Figure 5M

*Response: Thanks for the concern. Please consider that first, we have performed luciferase assays with **four different viruses** (Figure 5A-5D). Then, we have performed qRT-PCR with six different ISG's using CHIKV. After that, we performed western blot with 5 different ISG's with IRGM transient and stable knockdown HT29 cells. All of these layers of experiments provided us the same results. Therefore, we do not have any doubt about our data.*

*Nevertheless, as per the reviewer's suggestion, and to be more sure, we repeated the experiment of Fig. 5L using JEV (Please see **Figure 5M**). The data exactly mirrors the results obtained with CHIKV.*

Thanks for the suggestion and for helping us in making the data and conclusions more robust.

3. The results shown for APOBEC3 in Fig.2B and Fig.5J are highly different. Is this related to differences in the responsiveness of the shRNA HT-29 cells and the siRNA treatment? Again, this adds to the confusion of the differed cellular models used. The authors need at least to comment on this and explain why different cellular models were used.

*Response: We are sorry for the confusion. However, we didn't find any discrepancy in the indicated figures. Figure 2B depicts a heatmap of RNA sequencing analysis of **unstimulated (Basal) control and IRGM knockdown cells** where Figure 5J shows qRT-PCR with **uninfected and virus-infected control and IRGM knockdown cells**. These are two completely different*

conditions and experiments.

4. The work relies on the use of reporter cell lines and mRNA analysis to measure IFN induction. Could the authors could measure secreted type I interferon in selected critical assays and for the in vivo work (Fig. 4H,I; Fig. 5 A-F, Fig.5O)?

Response: Thanks for the concern. Measuring the ISGs levels is the standard method for assessing modulation of type 1 IFN response. In our study, for measuring IFN response, we have performed reporter assay, qRT-PCR, and western blotting with numerous ISG's. Now, we have measured the levels of IFN-β from the serum of IRGM wild type and KO mice (n=3, each group). The data show that IFN-β levels were significantly higher in IRGM KO mice serum (Supplementary Fig EV2C). Thanks for the suggestion.

5. As small samples sizes are used individual data points should be plotted in the graphs.

Response: Thanks for the suggestion. The graphs in Figure 1 and 2 has been changed to show individual points and all the new graphs made during revision are depicting data points.

6. Fig. 4D: Kaplan-Meier estimator should be used for statistical analysis and data shown as a Kaplan-Maier plot.

Response: Thanks for the suggestion. We have changed the graph accordingly.

7. Fig. 5 G-I: The authors nicely show that several MHC class I genes are regulated. It was shown that all of these are targets of NLRC5 which is a main regulator of MHC class I genes and an interferon target gene. It is thus likely that NLRC5 expression is the cause of the observed phenotype. Could the authors analyse NLRC5 mRNA expression in their samples or at least comment on this in the text.

Response: The reviewer is correct that NLRC5 is an important regulator of MHC class I gene expression and IRGM does regulate the expression of the IFN-regulated NLRC5 gene. This data we have shown in Figure 1E of our previous publication (Jena et al., 2020; EMBO reports). We have referenced the paper and commented on this now in the text (lines 183 -185). Thanks for the suggestion!

8. Interferon responses can affect cell differentiation. As the authors used differentiation protocols for THP1 cells and BMDMs this could be a cofounder. Could the authors show that IRGM/Irgm1 deletion/depletion does not affect macrophage differentiation?

Response: We routinely perform siRNA knockdown of IRGM in THP1 monocytes. We never observed any phenotypic change (even not minor) in these cell lines such as attachment to substratum upon IRGM knockdown. In addition, as mentioned above that the phenotypes are not just observed in THP-1 or BMDMs', most or all phenotypes are observed in several epithelial cells where no differentiation is required.

9. Polymorphisms in IRGM and its promotor are linked to autoimmune disease and Crohn's disease and the induction of type I interferon repones by IRGM loss-of function might be a key

event here. This should be mentioned as a word of caution when proposing targeting of IRGM a strategy for antiviral therapy.

Response: We agree with the reviewer and we have discussed this. Please refer to line number 426-438

Minor points:

- Description of the statistical analysis and software is missing in the material methods.

We have described the statistical analysis in each figure legend and now we have mentioned the software used in material and methods.

- Lines 132-148: The description of the ISG functions is rather long and in view of this reviewer not of help for the reader. The authors might reduce this part and add more discussion on the cell types used. Along these lines: Fig.2: Personally, this reviewer finds the panels A and F not very helpful, and they take a lot of space.

We have reduced the text in the mentioned paragraph. Fig 2A provides an overview of the genes (red marked) activated by IRGM and we think it is important to provide an overall picture of processes or mechanisms that are perturbed in IRGM depleted cells. We have now moved Figure 2F to supplementary. Thanks for the suggestion.

- Line 224: Remove statement on plaque assay as this is evident.

We have removed the statement. Thanks.

- Line 229: Typo "evaluated".

We have corrected it. Thanks.

- Lines 325-326: The indicated panels of Fig.4 are mixed up.

We have corrected it. Thanks.

- Line 337: "...the type I IFN response...."

We have corrected it. Thanks.

- Fig. 4C: It would help to show all the different scores instead of a compound score.

We think the graph with individual scores will not be able to show data clearly. The snapshot of excel used to generate this graph is shown on right.

- Fig. 5N: Please label the mRNA targets on the graph axes for clarity.

We have corrected it. Thanks.

	day 1	day 2	day 3	day 4	day 5
WT Mice-1	0	1	1	2	3
WT Mice-2	0	0	1	2	3
WT Mice-3	0	0	0	1	1
WT Mice-4	0	0	1	1	1
WT Mice-5	0	0	1	1	1
WT Mice-6	0	0	0	1	1
WT Mice-7	0	0	0	0	0
WT Mice-8	0	0	1	2	2
WT Mice-9	0	0	0	1	2
WT Mice-10	0	0	0	1	1
WT Mice-11	0	0	0	1	2
WT Mice-12	0	0	0	0	0
Score Sum	0	1	5	13	17
KO Mice-1	0	0	0	0	0
KO Mice-2	0	0	0	0	0
KO Mice-3	0	0	0	0	0
KO Mice-4	0	0	0	0	0
KO Mice-5	0	0	0	0	0
KO Mice-6	0	0	0	0	0
KO Mice-7	0	0	0	0	0
KO Mice-8	0	0	0	0	0
KO Mice-9	0	0	0	0	0
KO Mice-10	0	0	0	0	0
KO Mice-11	0	0	0	0	0
KO Mice-12	0	0	0	0	0
Score Sum	0	0	0	0	0

Referee #2:

This manuscript is a follow-up to a 2020 EMBO Reports paper from this group in which they demonstrated that IRGM1 interacts with cellular nucleic acid sensing proteins to direct them to the autophagy pathway to thus limit spontaneous interferon production. Here they confirm many aspects of their prior publication, and show that spontaneous interferon produced when IRGM1 is knocked down inhibits replication of many viruses in vitro. They further show that IRGM1^{-/-} mice are resistant to Chikungunya virus in vivo. In tangentially related experiments, they also show that antigen processing pathways are also increased when IRGM1 is deficient. Conceptually and mechanistically, this paper does not offer much of an advance beyond their previous paper, but it does contain an abundance of experiments and data that are well done and convincing.

We are very thankful to the reviewer for evaluating our manuscript and also praising the quality and amount of work.

We agree with the reviewer that in this manuscript we have taken forward the work published from our group (Jena et al., 2020 EMBO Reports). However, most humbly, we disagree that the data presented in this manuscript is just confirmatory and no conceptual advances are made. Please allow us to put forward the details that argue against this notion:

Figure 1. We presented data showing that IRGM protein expression is increased upon exposure to several viral PAMPs and viruses. To our knowledge, this is a piece of new and important information and has not been shown by any studies yet. Further, we showed that IRGM negatively regulates viral PAMPs induced IFN response. Before establishing the role of IRGM in antiviral response, it's important to establish that how IRGM is regulated and how it regulates ISG's in response to viruses and viral PAMPs.

Figure 2. In the first part of figure 2, we demonstrated that IRGM controls a large number of sentinel viral restriction factors. This is not the case with several negative regulators of IFN response reported in the literature. It is indeed an unprecedented observation as agreed upon by other reviewers also, which is not shown before and is a conceptual advance in understanding IRGM function.

In the second part of the figure, we demonstrated that IRGM knockdown or knockout cells are highly efficient in antigen processing and presentation. It is never demonstrated before that blocking IRGM expression could induce this important pathway. This work has implications not only in host-pathogen interaction but also in the cancer immunology field.

Figure 3. We found that downregulation of IRGM indeed establishes an antiviral state in the cells that can protect from a large number of viruses from five different families. IRGM manipulation can protect the host from some of the dreaded viruses including CHIKV, JEV, WNV, HSV-1, ZIKV, and SARS-CoV2. We believe that this is a sizeable demonstration of a broad anti-viral response to dreaded viruses and provide a conceptual advance in the field.

Figure 4. We presented data that shows that IRGM knockout mice are protected from CHIKV infection, whereas most wild-type (n=12) mice have succumbed to death. This is never shown before. And this validates our in-vitro data, which is important to establish IRGM as an ideal target for host anti-viral response

Figure 5. This figure shows that viral infected IRGM depleted cells have blunted IFN response. This is an important point for targeting IRGM for the antiviral response as IRGM depleted cells could restrict virus without producing excess inflammation.

Taken together, our work significantly advances knowledge in the field. It defines a new function of IRGM in host-virus interaction. This will be the first study to convincingly demonstrate that IRGM could be a good therapeutic target for enhancing a broad anti-viral immunity.

1. My only major concern with the data is that this group claims to have infected THP1 cells with SARS-CoV-2 when no other group in the world (to my knowledge) has been able to do so.

Response: A large number of studies show efficient infection of SARS-CoV and other coronaviruses in THP-1 and other macrophages (PMID: 15357874; PMID: 16501078; PMID: 17669539). We have performed SARS-CoV2 infection in THP-1 several times and in our hands, we never found problems. SARS-CoV2 viral load in THP-1 is high and the Ct cycle usually ranges between 20-22. When we compared viral load in THP1 cells at 10 hpi and 20 hpi, we found that the viral load is increased by ~8 folds (Fig EV3H) in 10 hours, which is possible only if there is efficient infection and transcription of the viral genome. This data is supported by a recent publication where they found that monocytes and macrophages (including THP-1) are efficiently infected by the SARS-CoV2 virus without a cytopathic effect. They also found that both undifferentiated and differentiated THP-1 cells have a similar viral load (Boumaza et al., 2021 J Infect Dis. PMID: 33493287. Figure 3 panel C). To further confirm this data, we performed plaques assays to understand whether the infection is productive. The supernatant from 24 hpi cells produced plaques however, they were found to be small and less in number (Fig EV3I and Fig 3S) indicating that productive virions are less efficiently produced in THP1 cells. Thanks for the suggestions and the same is now discussed in the manuscript text in lines 290 to 299.

Referee #3:

The work by Nath et al. displays that the reduced expression of IRGM in human cells, as well as the complete extinction of the related mouse Irgm1 protein, confer a very potent antiviral status to human cells/mice, respectively. IRGM deficiency correlates with the overexpression of multiple innate and adaptative immunity-related proteins and functions. As a consequence infection by several RNA and DNA viruses are much better controlled in cells with low IRGM expression. Additionally, Irgm1 KO neonates are highly resistant to CHIK virus infection, by contrast to their wt counterpart. The effect of the deficiency of IRGM on immune functions is impressive. As written by the authors "it is an unprecedented observation that so many of the key viral restriction factors can be induced by manipulation of a single gene".

Response: We are very thankful to the reviewer for reading our manuscript carefully and praising the work. We have now further improved the manuscript by incorporating the suggestion made by you and all the reviewers. Thanks a lot!

Some points could be done and or clarified

KO/Rescue experiments could be performed to display the specificity of the si/shIRGM treatments.

Response: We have now performed rescue experiments to establish the specificity of the IRGM mediated regulation of anti-viral response and viral infection. The results clearly show that overexpression of IRGM in IRGM knockdown cells rescues the phenotypes of IFN response and

viral replication (Figure 2F, Fig EV2E, and Fig EV3C). Thanks for this wonderful suggestion!

Fig 1 A-F complete western blot should be shown as supplementaries (not just the cutted bands).

Response: As per the suggestion of reviewers 1 and 3, we have repeated all the experiments in Figures 1A to 1F to show the kinetics of PAMPs treatment (compared to a single time point in the previous version of the manuscript). All the original blots shown in the manuscript are now provided separately as supplementary data as per the requirement of EMBO reports Journal.

IRGM reduction has already been reported to affect replications of several viruses. This work has to be referenced and discussed (PloSPathogens, 2011, 7, 12).

Response: We have now referenced and discussed the publication (Lines 410-422). We are very sorry for this mistake. The referenced paper identified that IRGM interacts with several viral proteins and proposed that autophagy defects in IRGM knockdown cells could be a possible reason for lower viral replication. The mechanism was not clear in this work. We don't refute their autophagy point but we have provided here strong evidence that activation of a large number of anti-viral mechanisms such as upregulation of several viral restriction factors, antigen processing, and presentation machinery, and stress granules upregulation in IRGM depleted cells are the direct reasons for defective viral replication in these cells. In addition, here we have shown that how IRGM is regulated by viruses/PAMPs and how it regulates virus/PAMPs induced IFN response. Further, we showed in vivo model that IRGM depletion could control viral replication very efficiently. Furthermore, we demonstrated that targeting IRGM could be useful in restricting several viruses of concerns including SARS-CoV2, ZIKA, CHIKV, and WNV. So these two works are very different from each other nevertheless, together they provide strong evidence that indeed IRGM is a robust therapeutic target for a broad anti-viral response. This is now discussed (Lines 410-422).

Since IRGM can be detected by western blot, protein expression has to be shown in siIRGM and shIRGM-treated cells (not only mRNA fold changes).

Response: As per the suggestion of the reviewer, we have now provided western blots of IRGM knockdown efficiency in all the blots (Please refer to figure 2E, 2F, 2P, 5K, 5L, 5M, 5N, and 5P).

Fig 1L: Type I IFN response is regulated through different pathways by viral PAMPs and by IFN- β . How the authors explained that both signals increased type I IFN response in siIRGM cells ?

Response: The reviewer is correct that different pathways are engaged to relay viral PAMPs

signaling for the production of type 1 IFN (IFN α and IFN β). These pathways include RIG-I-MAVS, cGAS-STING, and TLR3-TRIF. The type 1 IFNs thus produced induces JAK-STAT1/2 signaling in a paracrine and autocrine manner to enhance ISGs transcription (IFN response). In our previous publication (Jena et al., 2020, EMBO Reports) we have shown that IRGM regulates RIG-I-MAVS, cGAS-STING, and TLR3-TRIF pathways to regulate IFN β production and also regulate JAK-STAT1/2 signaling to regulate IFN response (Jena et al., 2020, EMBO Reports). This is the reason why both viral PAMPs and IFN- β could induce type 1 IFN response in IRGM depleted cells.

For a reader, it is not easy to understand how one single protein can as efficiently impact processes as different than internalisation / degradation / antigen presentation and type I IFN-related antiviral responses. Does the authors found one immune-related function not altered by the absence of IRGM ? Since many of the process analysed were reported to be induced by oxidative stress the oxidative status of IRGM defective cells should be looked at.

Response: In our previous study (Jena et al., 2020; EMBO Reports; PMID: 32715615), we showed that depletion of IRGM results in defective autophagy/mitophagy leading to the accumulation of defunct mitochondria. We also showed that because of the accumulation of these defective mitochondria the mtROS levels and total ROS levels in IRGM knockdown and knockout cells are increased (Figure 5; Jena et al., 2020; EMBO Reports; PMID: 32715615). The treatment of cells with ROS quencher, N-acetyl cysteine (NAC) partly reduces the IFN response in IRGM KD cells (Figure EV5F, Jena et al., 2020; EMBO Reports). Therefore, the reviewer is correct that increased oxidative stress contributes to increased IFN response in IRGM depleted cells. This has been well demonstrated in our previous work.

To determine the role of oxidative stress in viral replication, we treated the IRGM depleted cells with ROS quencher N-acetyl cysteine and then infected them with the VSV-eGFP virus. The NAC treatment of IRGM depleted cells significantly rescued the viral replication defect however was not able to completely restore it (Fig EV3J). Therefore, it appears that ROS play role in modulating immune responses and viral replication in IRGM depleted cells but it's not the sole reason.

This is correct that IRGM regulates several immune regulatory pathways. We and others have shown that IRGM negatively regulates NLRP3 inflammasomes, IFN response (cGAS, TLR3, TLR9, RIG-I), NF- κ B, and also p38-MAPK signaling (Mehto et al., 2019, Molecular Cell, Jena et al., 2020, EMBO Reports; Rai et al., 2021 Nature Immunology; Chauhan et al., 2015, Molecular Cell; Bafica et al., 2007, Journal of Immunology). However, the effect appears to be specific as it does not control AIM2 and NLRC4 inflammasomes (Mehto et al., 2019, Molecular Cell). Also, it appears it does not control MDA5 or TLR4-dependent signaling (Jena et al., 2020, EMBO Reports).

Since viral replication is affected by the absence of IRGM, we could ask whether the expression of the surface receptors of the tested viruses are as efficiently expressed on IRGM+ and IRGM- cells ? (What could impact virus entry)

We found that expression of ACE2 (SARS-CoV2 receptor) was almost similar in control and IRGM knockdown cells in two different cell lines tested (Fig EV3I) suggesting that perturbation of cell surface receptor may not be the prime reason for reduced viral load in IRGM depleted cells.

The reason of the supplementary fig 3C/D/E is not obvious and unnecessary.

The data indicate that depletion of IRGM in basal conditions induces host responses similar to the SARS-CoV2 infection. This data signifies that if we can generate IFN host response in the cell before infection then probably SARS-CoV2 infection can be suppressed as we have shown in the IRGM knockdown model. We believe that this

is an important point for future studies in the field. If the reviewer still thinks that it is unnecessary then we will be happy to remove it from the manuscript.

Do IRGM1^{-/-} mice have a higher titer of seric type I IFN than wt mice at steady state ?

Response: We have performed the experiment.

IRGM KO mice serum has higher levels of type 1 IFN (IFN- β) than wild-type mice. Please see Fig EV2C (n=3, each group) in the revised manuscript.

Overexpression of IRGM could be tested for its impact on antiviral gene expression, and virus replication.

Response: As per the reviewer's suggestion, we have now overexpressed IRGM and determined its effect on antiviral gene expression (Fig EV2D) and viral replication (Fig EV3K). Thanks for the suggestion!!

Referee #4:

Nath et al. report findings that extend their previous discoveries about the regulation of innate sensing by IRGM. Here, they show that loss of IRGM confers cells with resistance to infection by a diverse set of viruses. They associate this resistance with a high degree of interferon-stimulated gene induction when IRGM is silenced or knocked out. The authors include a huge amount of data in this manuscript, which is appreciated, but it seems some conclusions are premature. Overall, the relevance of their findings to human health remains uncertain because IRGM is an important negative regulator of spontaneous and infection-induced innate immunity, including inflammation.

Response: We are very thankful to the reviewer for carefully evaluating the work and praising it. We have now further improved the manuscript as per the suggestion made by reviewers. Thanks a lot!

In the context of human health relevance, we envision that IRGM inhibitors may be useful as broad anti-viral therapeutics including against SARS-CoV2. Our preliminary in-vitro data where we tested IRGM small molecule inhibitors shown us encouraging results (another study currently underway). We are currently in process of validating these inhibitors in vivo.

However, the reviewer apprehension in this context is very correct. Since IRGM is a negative regulator of inflammation, inhibition of IRGM may predispose a person to autoinflammatory conditions. However, we would like to argue that the use of such therapeutic option would be for a very short duration and probably as a prophylactic treatment, especially during epidemic or pandemic conditions. We think that usage of such therapeutics for a short duration probably may not have profound side effects and may be benefits will outweigh the side effects as true for most of the approved drugs. As suggested by the reviewer, we have now discussed this trade-off in the manuscript also. Line number 410 to 422.

Major:

Figure 1: The authors show time-course experiments using viral PAMPs indicating that IRGM is induced over time. However, the authors only show the effect of virus infection on IRGM at a single time point. Considering that the timing of IRGM induction is important to our understanding of how cells regulate the induction of antiviral ISGs, this is a crucial experiment.

Response: We performed the experiment as suggested by the reviewer. Thanks a lot for the suggestion!!

Figure 5: It is strange that no Viperin mRNA is detected in panel H, while protein is detected in panel L, and that protein level is increased upon IRGM shRNA.

Do these data suggest that IRGM can impact not only mRNA levels of ISGs but also protein levels (by affecting ISG protein turnover)?

Response: Please consider that Viperin (panel H), due to a huge increase in mRNA levels (~1700 folds) upon infection, Y-axis is in increment of 500, whereas it's not that case for the other targets (Figure 5G, I, J). Due to this reason, the values in the mock condition appear to be negligible in the previous graph (in the old manuscript), and also it appears that IRGM knockdown didn't change the mRNA levels of Viperin but it's not the case (Please See new graph in panel H and on right here). The mRNA levels of Viperin are significantly increased upon IRGM depletion in basal levels. So we don't think there is a discrepancy in mRNA vs protein data. This was our fault in the presentation of data. We are very sorry, and it is now corrected.

Nevertheless, we cannot completely rule out the possibility of IRGM-mediated turnover of Viperin directly at protein level via autophagy. Which could be another study itself.

Figure 5H

Line 351: The language used by authors is unclear and so I am unsure of whether the authors believe that virus replication is necessary in order for ISGs to be induced in this experiment. Furthermore, there are differential effects on the ISGs examined, suggesting that a blanket statement is inappropriate here. For example, Viperin protein levels are clearly enhanced by replicating virus and hardly at all by heat killed virus, while IFITM3 protein levels are induced by both replicating virus and heat killed. These data seem to suggest that certain ISGs are induced following replication while others are induced by PAMPs present in heat killed virus.

Response: We are very sorry for the confusion and for not a proper discussion of the results. Now we have discussed the results (Figure 5M) properly. We agree that the use of a blanket statement was inappropriate so now we have modulated the statements according to the results of each of the ISG's. Please refer to lines 367 to 376.

Here is the explanation why we performed the experiment in Figure 5 M:

Reason, hypothesis, and Results:

IRGM is a negative regulator of IFN response in basal conditions. Here, in this study, we found that upon exposure to viral PAMPs, the IRGM depleted cells showed higher IFN response than control cells. This suggests that even in stimulated conditions, IRGM negatively regulates IFN response. However, to our surprise, when we infected IRGM depleted cells with real viruses (JEV, CHIKV, or HSV), the IFN response was blunted in these cells. We hypothesized that this reduced level of ISG's in IRGM depleted cells could be due to reduced levels of replication of virus resulting in reduced levels of PAMPs for ISG induction. To validate this point, we performed the experiment

in Figure 5 M (new manuscript Figure 5N), where we infected cells with heat-killed viruses (lane 3 and lane 4) or transfected the cells with viral RNA isolated from CHIKV. In both conditions, we found that ISG's levels were not reduced in IRGM depleted cells (as compared to what we seen with alive viruses, Figure 5L and 5M) rather increased at least with viral RNA transfection. This partly proves our point that IRGM KD cells are highly resistant to virus replication resulting in lower levels of PAMPs in the environment leading to blunted IFN response. This is now discussed in the manuscript. Please refer to lines 367 to 376.

In this combined Results and Discussion format, the authors don't have the opportunity to discuss the trade-offs of targeting IRGM as a host-directed therapy. For example, how would the authors plan to minimize autoimmune-related disease resulting from unchecked interferon production, in the absence and presence of virus infection?

Response: We apologize for not discussing the trade-off of targeting IRGM as a host-directed therapy. We have now discussed this. Please refer to lines 410 to 422.

The authors do not discuss how the GTPase function of IRGM contributes to its regulation of ISGs.

Response: Thanks for asking this important question. Rather than just discussing, we have now performed experiment to show the role of GTPase activity in the regulation of ISGs. The GTPase activity of IRGM is important for its autophagic and anti-inflammatory function and a single

Figure 2G

mutation (S47N) in GTPase domain renders it inactive (Jena et al., 2020, EMBO Reports; Mehto et al., 2019, Mol. Cell; Kumar et al., 2018, JCB; Singh et al., 2010, Nature Cell Biology). We found that compared to wild-type IRGM, the S47N catalytic (GTPase) mutant of IRGM was not able to suppress IFN response efficiently (Figure 2G). Thanks for the suggestion!!

Fig EV2H

Can the authors also test how BMDM take up transferrin-AF 488 and LDL-AF 488? That way, the authors could distinguish between an effect of IRGM on endocytic activity versus a selective effect on ovalbumin-AF 488 uptake.

Figure 2: Can the authors also test how BMDM take up transferrin-AF 488 and LDL-AF 488? That way, the authors could distinguish between an effect of IRGM on endocytic activity versus a selective effect on ovalbumin-AF 488 uptake.

Response: The uptake of both soluble ovalbumin and transferrin is dependent upon receptor-mediated endocytosis (PMID: 20308535, PMID: 16709836, PMID: 2396980). The endocytosis of ovalbumin takes place through the mannose receptor whereas transferrin is taken up by the transferrin receptor. We performed the experiment as suggested by the reviewer. The depletion of IRGM results in enhanced uptake of transferrin-AF 488 (Fig EV2H). The data suggest that IRGM depletion induces endocytosis (in general) and is not specific for ovalbumin. This is now discussed in the manuscript.

Throughout the figures, IRGM immunoblots are not shown following IRGM silencing or knockout, for some reason.

Response: We are sorry for our oversight, IRGM immunoblots are now provided with all western blots. (Please refer to figure 2E, 2F, 2P, 5K, 5L, 5M, 5N, and 5P).

Minor:

Put lines around western blot boxes for increased readability.

Response: We have put boxes around western blots.

Figure 1T: One of the columns is white when it should be black.

Response: We are sorry, we have corrected this.

Figure 3: the term "viral load" is inappropriate when referring to the extent of virus infection detected in cell culture. Instead, the authors should use "% Envelope+ cells" and "% Nucleocapsid+ cells" on the Y axes.

Response: We have corrected this. Thanks for the suggestion!

—

Dear Dr. Chauhan,

Thank you for the submission of your revised manuscript to our editorial offices. I have now received the reports from the four referees that were asked to re-evaluate your study, you will find below. As you will see, the referees now fully support the publication of your study.

Before we can proceed with formal acceptance, I have these editorial requests I ask you to address in a final revised version of the manuscript:

- I would suggest a slightly modified title:

Inhibition of IRGM establishes a robust antiviral immune state to restrict pathogenic viruses

- Please provide the abstract written in present tense.

- Please restrict the keywords on the title page to 5.

- Please have your final manuscript text carefully proofread by a native speaker. There are still several grammatical errors present (e.g. regarding the use of 'mice' and 'mouse').

- Please make sure that all the funding information is entered into the online submission system and is complete and similar to the one in the manuscript text file.

- Please name the movies 'Movie EV#' and also use this name as callout in the manuscript text. Please upload the movies ZIPed together with a text file containing their title and their legend. Finally, please remove the movie legends from the main manuscript text.

- Please name the legends for the EV figures in the manuscript text 'Expanded View Figure Legends'.

- Please make sure that the number "n" for how many independent experiments were performed, their nature (biological or technical replicates), the bars and error bars (e.g. SEM, SD) and the test used to calculate p-values is indicated in the respective figure legends (also of the EV figures), and that statistical testing has been done where applicable. Please avoid phrases like 'independent experiment', but clearly state if these were biological or technical replicates. If statistical testing was done but there is no significant difference, please also mark this in the diagrams (n.s.).

- Please add scale bars of similar style and thickness to all the microscopic images, using clearly visible black or white bars (depending on the background). Presently, some of the bars are too thin (see e.g. Fig. EV3D). Please place these in the lower right or left corner of the images. Please do not write on or near the bars in the image but define the size in the respective figure legend.

- Thank you for providing the source data for the Western blots. Could you please separate these and submit one PDF file per figure? For the source data for the third EV figure please use the name Figure EV3 and check the panel (it seems to be panel J not I).

- In the author contributions two authors are mentioned as 'SC'. Please distinguish these, e.g. by using 'SChau' and 'SChat'.

- Finally, please find attached a word file of the manuscript text (provided by our publisher) with a few changes and queries we ask you to include in your final manuscript text. Please provide your final manuscript file with track changes, in order that we can see any modifications done.

In addition, I would need from you:

- a short, two-sentence summary of the manuscript (not more than 35 words).
- two to four bullet points highlighting the key findings of your study.
- a schematic summary figure (in jpeg or tiff format with the exact width of 550 pixels and a height of not more than 400 pixels) that can be used as a visual synopsis on our website.

Best,

Achim Breiling
Editor
EMBO Reports

Referee #1:

The authors have carefully and thoroughly addressed all my queries. The provided novel data and data analysis significantly improves the impact of the manuscript and confirm the findings of the previous experiments.
I recommend publication of the study.

Referee #2:

My previous comments were minor and have been addressed. The comments of the other reviewers have also been largely addressed.

Referee #3:

The authors have addressed most of the issues raised in their original submission.

Referee #4:

The authors have fully addressed my questions and suggestions.

The authors have addressed all minor editorial requests.

Dr. Santosh Chauhan
Institute of Life Sciences
Cell Biology
Nalco square rd
Chandrasekharpur
Bhubaneswar, Odisha 751023
India

Dear Dr. Chauhan,

I am very pleased to accept your manuscript for publication in the next available issue of EMBO reports. Thank you for your contribution to our journal.

At the end of this email I include important information about how to proceed. Please ensure that you take the time to read the information and complete and return the necessary forms to allow us to publish your manuscript as quickly as possible.

As part of the EMBO publication's Transparent Editorial Process, EMBO reports publishes online a Review Process File to accompany accepted manuscripts. As you are aware, this File will be published in conjunction with your paper and will include the referee reports, your point-by-point response and all pertinent correspondence relating to the manuscript.

If you do NOT want this File to be published, please inform the editorial office within 2 days, if you have not done so already, otherwise the File will be published by default [contact: emboreports@embo.org]. If you do opt out, the Review Process File link will point to the following statement: "No Review Process File is available with this article, as the authors have chosen not to make the review process public in this case."

Should you be planning a Press Release on your article, please get in contact with emboreports@wiley.com as early as possible, in order to coordinate publication and release dates.

Thank you again for your contribution to EMBO reports and congratulations on a successful publication. Please consider us again in the future for your most exciting work.

Yours sincerely,

Achim Breiling
Editor
EMBO Reports

THINGS TO DO NOW:

You will receive proofs by e-mail approximately 2-3 weeks after all relevant files have been sent to our Production Office; you should return your corrections within 2 days of receiving the proofs.

Please inform us if there is likely to be any difficulty in reaching you at the above address at that time. Failure to meet our deadlines may result in a delay of publication, or publication without your corrections.

All further communications concerning your paper should quote reference number EMBOR-2021-52948V3 and be addressed to emboreports@wiley.com.

Should you be planning a Press Release on your article, please get in contact with emboreports@wiley.com as early as possible, in order to coordinate publication and release dates.

Corresponding Author Name: Dr. Santosh Chauhan

Journal Submitted to: EMBO REPORTS

Manuscript Number: EMBOR-2020-50051V1